# Non-Canonical Male Meiosis in a Marine Gastropod, *Littorina saxatilis*

**DOI:** 10.3390/biology14111572

**Published:** 2025-11-09

**Authors:** Sergei Iu. Demin, Natalia A. Mikhailova, Andrei I. Granovitch, Dmitry S. Bogolyubov

**Affiliations:** 1Institute of Cytology of the Russian Academy of Sciences, 194064 St. Petersburg, Russia; serghei.demin@gmail.com (S.I.D.); natmik@mail.ru (N.A.M.); 2Department of Invertebrate Zoology, Faculty of Biology, St. Petersburg State University, 199034 St. Petersburg, Russia; a.granovich@spbu.ru

**Keywords:** meiosis, diffuse stage, “fuzzy” chromosomes, karyosome, chiasmata, marine snails, *Littorina saxatilis*

## Abstract

**Simple Summary:**

The rough periwinkle *Littorina saxatilis* has become a model species for evolutionary biology in recent decades. The reproductive strategy of this species, including meiosis, is of particular importance. A cytogenetic study using nuclear spreads and live cell observations revealed for the first time some non-canonical features of male meiosis in this marine snail. The most intriguing are the diffuse stage and the karyosome stage during prophase I, as well as the delay in the appearance of chiasmata and chromosome-specific chromomeric patterns of bivalents during diakinesis—early anaphase I.

**Abstract:**

An atypical course of male meiosis in *Littorina saxatilis* from zygotene to early anaphase I has been established, which includes non-canonical stages—diffuse and karyosomal. In diakinesis, a structural stepwise transition of bivalents from a single-thread, homogeneously colored form to a double-thread and banded form was discovered. In early diakinesis, in addition to bivalents without pronounced chiasmata, which constitute the majority, rare cruciform short bivalents with one chiasma are revealed. In mid-diakinesis, two or three types of bivalents with one or two chiasmata and several achiasmatic bivalents are identified. In late diakinesis—metaphase I—seven types of bivalents were distinguished, bearing from one to three chiasmata. Some bivalents of the set showed noticeable chiasmata only in early anaphase I. Therefore, the course of diakinesis in *L. saxatilis* male meiosis is rather atypical. In pachytene, mid- and late diakinesis, and partly in metaphase I, individual bivalents of the *L. saxatilis* set are reliably identified because they exhibit chromomeric patterns similar to those of the G-banded prometaphase chromosomes of early embryos and spermatogonial mitotic cells. Our research provides a cytological basis for further studies of conservation/variability and evolution of male meiosis.

## 1. Introduction

Meiosis is the fundamental process of sexual reproduction leading to the formation of haploid gametes from a diploid progenitor cell, which involves one round of DNA replication followed by two successive rounds of chromosome segregation. Pairing and segregation of homologous chromosomes are two fundamental features of the meiotic program [1,2]. Without cytological and genetic data on the canonical and non-canonical meiosis programs that can be implemented in different organisms, it is impossible to understand the evolution of eukaryotes [3]. However, our knowledge in this field is still limited to a few model species.

The typical (canonical) mode of meiosis has been characterized in detail and referred to as mainstream [4]. In this mode, during meiotic prophase, the germline cells (meiocytes) consecutively enter the stages of leptotene, zygotene, pachytene, diplotene, and diakinesis. In the period from the late leptotene to the end of zygotene, when synapsis of homologous chromosomes begins, all chromosomes are attached by their telomeres in a small region of the nuclear envelope, forming a conserved cluster called the meiotic bouquet [5,6]. Although the polarized arrangement of chromosomes—the bouquet—is highly conserved among eukaryotes, the morphological patterns of the classical bouquet are not always obvious, which may reflect meiotic evolution [7].

In pachytene, a bouquet arrangement of chromosomes disappears, and bivalents are formed with paired and fused chromomeres (G-bands) of homologous chromosomes and with always paired interchromomeres (R-bands) [1]. The segregation of homologous chromosomes in diplotene leads to an increase in the distance between homologues. This occurs due to the partial disassembly of synaptonemal complexes (SCs). At this stage, it is possible to estimate the number of crossover exchanges per bivalent by counting chiasmata—the cytological manifestations of crossing over [8]—which represent an X-shaped crossroad of recombinant nonsister chromatids. According to the classification of chiasmata [9], one chiasma in an interstitial fragment of acrocentic chromosome (or chromosome arm) constitutes an intrabivalent region with a cruciform configuration; two chiasmata establish a region with a ring-like configuration; and three chiasmata create a region with a joint (double-ring) configuration. In diakinesis, bivalents are strongly compacted.

At the same time, meiosis is a fairly flexible process in various eukaryotes, demonstrating some considerable variations in its course in different species. In particular, recombination does not necessarily occur after chromosome pairing. In many plants and animals, obvious chiasmata are not formed and crossing over does not occur [10]. It is assumed that the sporadic occurrence of achiasmatic meiosis among organisms is polyphyletic in nature and can be explained by the secondary loss of meiotic recombination during evolution [11]. The lack of chiasmata is mostly due to the absence of DNA double-strand breaks (DSBs), so that most achiasmatic organisms may rely on DSB-independent homology recognition mechanisms for chromosome pairing [12]. Only in few organisms—such as females of the silkworm *Bombyx mori* [13,14] or males of the scorpion *Tityus silvestris* [15]—the formation of DSBs occurs in an earlier stage of meiosis, which are repaired in a crossing over-independent manner [12].

To better understand the evolution of meiosis, it is necessary to know the mechanisms of formation and functions of those unique and specific forms of the chromosomal apparatus that can be formed during meiotic prophase. These primarily are the lampbrush chromosomes [16,17,18,19,20,21] and the karyosome—an evolutionarily conserved structure formed at the prolonged diplotene stage by condensed chromosomes assembled together in a limited space of the nucleus [22]. Both lampbrush chromosomes and the karyosome are more characteristic of female than male meiosis [23]. However, the non-canonical karyosome stage characterizes male rather than female meiosis of the nematode *Caenorhabditis elegans* [24]. In this species, the karyosome stage is an intermediate stage between typical diplotene and diakinesis. Earlier stages of meiotic prophase may be characterized by another non-canonical stage of meiosis I—the diffuse stage—when chromatin relaxes during the transition from pachytene to diplotene [25,26,27].

The rough periwinkle *Littorina saxatilis* (Mollusca, Gastropoda) and its closely related species represent a good model to study the mechanisms of speciation and reproductive isolation in the context of the key problems of evolutionary biology. Some studies have mainly focused on *Littorina* genomics [28,29], transcriptomics [30], and proteomics [31]. Progress has also been achieved in understanding the molecular mechanisms of gametic incompatibility [32] and the contribution of gene flow and natural selection to the diversification of *Littorina* sibling species [33]. Surprisingly, however, the features of spermatogenesis and meiosis in *Littorina* remain poorly understood [34].

We performed a cytological study to characterize male meiosis of *L. saxatilis* and unexpectedly found that its progression from zygotene to early anaphase I is atypical. Most importantly, male meiosis of *L. saxatilis* exhibited non-canonical diffuse and karyosome stages and had an unusual delay in chiasmata appearance at diakinesis. It was also surprising that individual bivalents of *L. saxatilis*—at pachytene, mid and late diakinesis, and partly at metaphase I—can be reliably identified, since their chromomeric patterns are very similar to those of G-banded prometaphase chromosomes of early embryos and spermatogonial mitotic cells.

## 2. Materials and Methods

Sexually mature specimens of *L. saxatilis* (Olivi, 1792) were collected in spring at Tromsø Island (Norway) in 2019. The mollusks were kept in the laboratory conditions as described [34]. The primary spermatocytes were observed in a camera for living observation [34] under a Leica DM 2500 microscope (Leica Microsystems, Wetzlar, Germany) equipped with Nomarski optics (DIC) and a DFC 420 CCD camera, using a 100× oil immersion objective (NA 1.25) (Leica Microsystems, Wetzlar, Germany). The study was approved by the Animal Ethics Committee of the Institution of Cytology of the Russian Academy of Sciences (approval date: 10 March 2023, protocol # 18/23).

Preparations of spread spermatocytes and oocytes were obtained using the high-pressure technique developed by us earlier [35]. The gonads were incubated in hypotonic water solution (130 mM KCl) for 20 min at room temperature (RT), then fixed with absolute methanol and glacial acetic acid (3:1) for 2–4 hrs at RT and stored at −20 °C before use. The fixed gonads were divided into small pieces in 50% propionic acid, then covered with a siliconized coverslip (24 × 24 mm), after which the slides were placed in a vertical hydraulic vise equipped with a manometer, and then pressure was gradually increased to about 250 kg/cm^2^ for 90–120 s. The slides were frozen in liquid nitrogen, after that the coverslip was removed with a razor blade, and then the preparations were dehydrated in an ascending series of ethanol (70, 80, and 100%), dried in air, and stored at −20 °C.

The cellular composition of the testes was determined in semithin sections of 0.5 μm thick, stained with toluidine blue. The testes were fixed in 2.5% glutaraldehyde in 0.05 M cacodylate buffer (pH 7.2) and then in 1% osmium tetroxide in the same buffer. After dehydration in an ascending ethanol series, the specimens were embedded in Epon 812 (Fluka Chemie AG, Buchs, Switzerland).

For cytological analysis, the preparations were stained with silver nitrate according to the standard AgNOR procedure [36]. We also used a routine chromatin staining technique with 5% Giemsa solution in Gurr buffer, pH 6.8. One hundred and fifty cells were analyzed for each stage of male meiosis and staining procedure, with the exception of rare spermatogonial, zygotene, anaphase I and II cells (*n* = 50 for each of these stages).

For karyotypical analysis, the preparations were pretreated with RNase A and collagenase according to the methods described previously [37,38]. The slides were incubated in RNase A solution (100 μg/mL) in 2× saline-sodium citrate (SSC) buffer (Sigma-Aldrich, Burlington, MA, USA) during 1 h at 37 °C, and then washed twice in 2× SSC for 5 min. The preparations were also treated with 1 mg/mL collagenase in PBS (Sigma-Aldrich, Burlington, MA, USA) for 7 min at RT and rinsed 3 × 5 min in PBS. DNA was stained with 0.5 mg/mL AT-specific fluorochrome DAPI (Sigma-Aldrich, Burlington, MA, USA) in McIlwaine’s buffer (pH 7.0) for 15 min at RT. The preparations were embedded into a ProLong^®^ Gold antifade reagent (Invitrogen, Waltham, MA, USA). Spreads of meiotic and mitotic cells were imaged with a Leica DMI 6000 inverted microscope (Leica Microsystems, Wetzlar, Germany) equipped with a Leica DFC350 FX digital camera. Initial DAPI-banded chromosomes images were enhanced with the Fast filters package of ImageJ software (v. 1.50i). To analyze the chiasmata in diakinesis –metaphase I, we used the classification proposed by Sidhu with co-workers [39]. Eighteen karyograms of primary spermatocytes at diakinesis–early metaphase I, twelve spermatogonial early metaphase karyograms, and ten early metaphase karyogram of early embryo cells—were analyzed for construction of a collection of high-resolution G-banded images of individual *L. saxatilis* chromosomes, termed the Image Bank in this study.

## 3. Results

The rough periwinkle *Littorina saxatilis* has a peculiar type of lobular-cystic spermatogenesis, rarely found in mollusks. In this case, male germline cells develop in multicellular cysts (spermatocysts) that fill each lobe of the testis [40]. The details of the structure of these spermatocysts are poorly distinguishable on conventional light microscopic sections, so each lobe of the testis appears filled with accumulations of germ cells at various stages of spermato- and spermiogenesis (Figure 1). The stages of spermatogenesis and spermiogenesis in *L. saxatilis* have been determined previously [34].

### 3.1. Sequential Stages of L. saxatilis Male Meiosis

The AgNOR staining method made it possible to identify spermatogonia, as well as primary and secondary spermatocytes at successive stages of meiosis, according to the state of their chromosomal-nucleolar apparatus (Figure 2a–y). Additional information was obtained from analyses of DAPI-stained freshly squashed (unfixed) cells and unfixed/unstained cells observed by differential interference contrast microscopy (DIC live imaging).

#### 3.1.1. Spermatogonia

Spermatocysts containing spermatogonia were rarely encountered in our material collected during the main breeding season of *L. saxatilis*. The nuclei of interphase spermatogonia are filled with diffuse chromatin and contain 3–4 large nucleoli (Figure 2a).

#### 3.1.2. Primary Spermatocytes

The nuclei of cells that we have classified as interphase primary spermatocytes are morphologically similar to the nuclei of spermatogonia, with chromatin diffusely distributed through the nucleus (Figure 2b). Three to four nucleoli are also present, but they were three to four times smaller than in spermatogonia (Figure 2b). In contrast to spermatogonia, interphase primary spermatocytes could be frequently found in the testes of all specimens studied.

##### Leptotene, Zygotene and the Stage of Interlock Resolution

The stages of leptotene and zygotene morphologically can be referred to as canonical stages of *L. saxatilis* male meiosis. At the leptotene stage, the spermatocyte nucleus contain two to three nucleoli and slightly entangled single-thread chromosomes (Figure 2c). In late leptotene—early zygotene spermatocytes we did not observe the classical chromosome bouquet [5,6,7]. However, clusters of condensed chromatin, apparently reflecting the onset of synapsis, were visible (Figure 2d), especially after DAPI staining (Appendix A). Therefore, we refer to such patterns as “bouquet-like.”

In the zygotene (Figure 2e), homologous chromomeres of the forming bivalents were observed paired or had bands due to cross-bridges between them. All nuclei contained two nucleoli, one or both of which were more than twice as large as in previous stages. The interlocking stage (the stage of interlock resolution) [41,42,43,44,45] occurs during the transition from zygotene to pachytene. It is characterized by the appearance of bivalents bearing extended areas of desynapsis of homologous chromosomes (Figure 3).

##### Pachytene

The pachytene stage in male meiosis of *L. saxatilis* can be divided into early, mid and late substages depending on the state of fusion of homologous chromomeres (Figure 2f–h, Appendix A). Early pachytene spermatocytes contain 17 bivalents (Figure 2f), the homologous chromomeres of which are connected by a cross-bridge, form a band, or, less commonly, completely fuse to form a single chromomere. In late pachytene spermatocytes, all bivalents appear as necklace-like structures due to the complete fusion of both homologous chromomeres (Figure 2h and Figure 4a). At this pachytene substage, no significant shortening of bivalents was observed compared to early pachytene (Figure 2h). Mid-pachytene spreads exhibited both banded and necklace-like bivalents (Figure 2g, banded bivalents are marked with arrows). All spermatocyte nuclei contained two Ag-positive nucleoli (Figure 2f–h).

##### Diffuse Stage

We found that in male meiosis of *L. saxatilis*, late pachytene is followed by a diffuse stage, when bivalents do not shorten, and homologous chromosomes are located even closer to each other, completely filling the nucleus. The chromomeric organization of bivalents becomes indistinct due to a strong decrease in the level of chromomere condensation. Short chromatin loops are formed at the periphery of the bivalents, which causes their characteristic “fuzzy” structure, somewhat reminiscent of the appearance of lampbrush chromosomes. (Figure 4b). One of the two nucleoli of the spermatocyte disappears at this stage (Figure 2i), so that only one nucleolus remains subsequently (Figure 2j–m).

##### Diplotene and Karyosome Stage

Another unexpected feature of male meiosis in *L. saxatilis* was the detection of atypical diplotene (Figure 2j), when bivalents exhibit a lampbrush-like, fuzzy structure due to chromatin loops at their periphery and DAPI-positive bead-like structures in the axial elements (Figure 5c).

Later on, a karyosome is formed in the nucleus of spermatocytes, which is another non-canonical feature of male meiosis in *L. saxatilis*. The karyosome stage, in turn, can be divided into the pre-karyosomal, complete karyosomal, and post-karyosomal substages (Figure 2k–m, Figure 4d–f and Appendix A). The pre-karysome, sometimes called the early karyosome/karyosphere [23], reflects the incomplete packaging of all chromosomes in the nucleus into a single “knot” (karyosome). At this substage, a certain chromosome-free space is visible in the nucleus. When the process of chromosome aggregation into a karyosome is complete, this chromatin structure occupies a rather limited space in the nucleus, which corresponds well to the definition of a karyosome [22,23]. Subsequently, the karyosome subsequently loosens again, and we called this period the post-karyosomal substage, preceding the onset of diakinesis.

In *L. saxatilis* spermatocytes, bivalents assembled into the karyosome retain their morphological individuality (Figure 2l and Figure 4e), in contrast to the karyosomes in female meiosis of a number of organisms [22]. At the post-karyosomal stage, 17 short bivalents without any signs of chromomeric organization are distinguishable (Figure 2m and Figure 4f). The *L. saxatilis* male karyosome was always observed in close proximity to the nuclear envelope (Appendix A).

##### Diakinesis

Based on the severity of internal cleavages, chiasmata, and G-band patterns of individual bivalents (see below), reflecting the progression of diakinesis, we were able to distinguish three morphological types of diakinetic nuclei (Figure 2n–p). In early diakinesis, one-third to one-half of the bivalents in the set contained interstitial splited filamentous regions flanked by chromomeric bead-like areas (Figure 2n). The interstitial fragments of the remaining bivalents had a constriction. Among bivalents without recognizable chiasmata, there were rare short cruciform bivalents with one chiasma (Figure 2n and Figure 5a).

In mid diakinesis, bivalents showed bands (Figure 2o). One or two of the shortest bivalents had a noticeable split between the two homologous chromosomes. We observed 2–4 types of bivalents with one or two chiasmata (Figure 5a–c,e) and several bivalents without them.

In late diakinesis, signs of fusion of homologous chromomere were weakly expressed (Figure 2p). Interchromosomal splits are now clearly visible in most bivalents. We were able to distinguish 7 types of bivalents, each bearing from one to three chiasmata (Figure 5a–g). The nucleolus organizer regions (NORs) were not detected at all stages of diakinesis (Figure 2n–p).

##### Metaphase—Telophase I

In early metaphase I, bivalents become shorter and narrower than during diakinesis (Figure 2q). As a rule, interchromosomal splits are not visible at this stage and appear only in late metaphase I (Figure 2r). While metaphase I progresses, the bivalents become even shortened and thinned (Figure 2q–s). In anaphase I (Figure 2s), homologous chromosomes are completely distributed into daughter haploid sets. At this stage, chromosomes reach their smallest size. NORs were not detected in metaphase and anaphase I spermatocytes and appeared again—one or two per nucleus—only at telophase I (Figure 2t).

#### 3.1.3. Secondary Spermatocytes

The nucleus of secondary spermatocytes can be identified by its size, which is half the size of the interphase nucleus of primary spermatocytes, and by the pattern of Ag-NOR staining. The interphase nucleus of secondary spermatocytes contain 3–7 Ag-positive zones (Figure 2u), represented by either a single argentophilic granule or a linear sequence of several granules. In prophase—metaphase II, NORs were not detected (Figure 2v–w). They appeared only during the transition from anaphase to telophase II (Figure 2x). The number of argentophilic zones in late telophase II spermatocytes was the same as in the nuclei of interphase secondary spermatocytes. The interphase and telophase nuclei of secondary spermatocytes differ in size: the latter are approximately three times smaller than the former. The nuclei of late telophase II spermatocytes (Figure 2y) showed similar pattern of AgNOR staining as interphase secondary spermatocytes (Figure 2u).

### 3.2. Comparison of the Pachytene—Diplotene Period in Male and Female Meiosis of L. saxatilis

In female meiosis, we found that the nucleus of primary oocytes has one large nucleolus and a typical bivalent organization (Figure 6a–c). All oocytes’ pachytene bivalents exhibit homologous chromomere pairs in the interstitial area, flanked by banded regions. The one-thread regions of bivalents were represented by a very small number of fused homologous chromomeres, no more than 1–4 per bivalent (Figure 6a). In diplotene oocytes, the length of bivalents was almost half their length at the pachytene stage. Homologs located collaterally were separated by pronounced interchromosomal splits, held together by terminal and interstitial chiasmata (Figure 6b,c). The number of interstitial chiasmata in individual bivalents varied from complete absence to 3 or even 4. Separation of homologous chromosomes in bivalents was observed in late diplotene. An increased level of chromatin condensation and even chiasmata terminalization, which is usually characteristic of diakinesis, were also registered (Figure 6c). The bulk of oocyte bivalents at the diplotene stage of female meiosis had banded regions separated by constrictions at the chiasmata.

In spermatocytes, on the contrary, pachytene bivalents (Figure 6d) are significantly shorter than in oocytes, most likely due to the intertwisting of homologous chromosomes in these bivalents. This intertwisting characterized all bivalents in the spermatocyte set, with the exception of 1–3 the smallest chromosomes, which showed only bands. In pachytene bivalents of spermatocytes, true chiasmata were not observed.

In male diplotene (Figure 6e), bivalents with lateral chromatin loops were observed, demonstrating laterally fused homologous chromomeres. The bivalents of diplotene spermatocytes (Figure 4e) appeared comparatively shorter and narrower than oocyte bivalents at this stage (Figure 4b). In early diakinesis (Figure 4f), the spermatocyte bivalents become even shorter and thinner compared to the previous stage (Figure 6e).

### 3.3. G-Banding and Chromomere/Interchromomere Patterns of Chromosomes During L. saxatilis Mitosis and Male Meiosis

#### 3.3.1. The Image Bank

We compiled a collection of high-resolution G-band images of individual *L. saxatilis* chromosomes, termed the Image Bank in this study, which was then used as a primary tool to identify individual bivalents in spreads and to compare G-banding pattern in bivalents at pachytene, diakinesis, and metaphase I of male meiosis in this species. The initial source for creating the Image Bank was images obtained from several dozen karyograms of G-banded chromosomes of spermatogonial and early mitotic cells of the embryo at the stages of prometaphase and early metaphase. For presentation of the Image Bank, gray images of individual G-banded chromosomes selected from the karyograms were enhanced and computer straightened. We further used images taken from the Image Bank as a standard for identifying individual chromosomes or bivalents.

We found that G-band patterns vary little among the individual chromosomes of embryonic and gonial cell sets at the same resolution (Figure 7). Individual chromosomes presented in the Image Bank differed from each other mainly in the degree of association/dissociation of sister chromatids.

#### 3.3.2. G-Banding Patterns of Individual Early and Late Pachytene Bivalents

Using the created Image Bank (Figure 7) and based on G-banding patterns, we compared the structure of pachytene bivalents in male meiosis of *L. saxatilis* with the structure of mitotic chromosomes in gonial and embryonic cells (Figure 8). Surprisingly, at early and late pachytene, individual bivalents of the set did not differ from each other in their chromomeric patterns. Moreover, early metaphase mitotic chromosomes did not differ in G-banding patterns—neither in the resolution of the chromomeric structure nor in the staining intensity and relative sizes of homologous chromomeres and the corresponding G-bands. In late pachytene of *L. saxatilis* male meiosis, all homologous chromomeres were observed fused in all individual bivalents of the set.

#### 3.3.3. Delayed Expression of Chiasmata and Chromomeric Patterns of Individual Bivalents During Late Diakinesis—Metaphase I

In late diakinesis—metaphase I, individual bivalents of the set demonstrated a chromomeric pattern that corresponded to the G-banding pattern of the least resolved early metaphase chromosomes presented in the Image Bank of individual chromosomes (Figure 7).

In late diakinesis (Figure 9), one group of individual bivalents of the set had the appearance characteristic of early pachytene and the same width, but noticeably shorter length. Another group contained a short zone of asynapsis at the centromeric position, affecting one chromomere flanked by two asynaptic interchromomeres. It is noteworthy that the chromomeres, heterochromatized in pachytene and mitosis and intensely stained with Giemsa dye, become greatly loosened upon entering the asynaptic zone in late diakinesis and stain much more weakly. The third group consisted of typical chiasmatic bivalents, the chromosomal ascription of which could be unambiguously determined from the chromomere pattern using a standard taken from the Image Bank.

On Giemsa-stained spreads of metaphase I cells (Figure 10), 14 bivalents were unambiguously identified using standards from the Image Bank (see above). Compared with diakinesis, due to a decrease in the resolution of the chromomeric pattern associated with a noticeable shortening of bivalents in metaphase I, three bivalents could not be distinguished based on their chromomeric pattern. In metaphase I, as in late diakinesis, a group of bivalents with delayed dissociation of homologous chromomeres was preserved, although this group included 4 bivalents (Figure 10). In metaphase I, chiasmata were clearly identified in a group comprising the 7 longest bivalents of the set, making their enumeration and chromosome mapping possible. In this case, only single homologous chromomeres of the bivalent, located in the chiasm zone, were in a state of fusion. Among the short bivalents of the set, the group in which 6 homologous chromosomes are connected by a long filamentous chiasm has increased (Figure 10). At this stage of meiosis, single dissociated homologous chromosomes without visible chromatin contacts were also encountered.

## 4. Discussion

### 4.1. Karyotype of Littorina saxatilis and the Image Bank of Individual Chromosomes

*L. saxatilis* specimens collected by us on the coast of Tromsø Island (Norway) showed a haploid chromosome number of *n* = 17. Other researchers have also recorded 2n = 34 for various populations of this species worldwide [46,47,48,49,50]. Heteromorphic bivalents were not detected in the karyotypes of *L. saxatilis* spermatocytes.

Our Image Bank—a collection of high-resolution images of *L. saxatilis* individual G-banded mitotic chromosomes—has become an important tool for assessing the structure of meiotic chromosomes in this species (Figure 7). It also appears suitable for the reliable identification of individual chromosomes/bivalents of a set in cells of different origins. We previously created a similar Image Bank for prophase chromosomes of the trematode *Himasthla elongata* [35].

Given the issue of sibling/cryptic species of the genus *Littorina*, it should be noted that the early embryos we used for karyotyping were isolated from the brood pouch characteristic of females of only one *Littorina* species—*L. saxatilis* [51]. In males, the reliability of morphological or molecular criteria for species diagnosis of various cryptic species of the genus *Littorina* does not exceed 70–80% [52]. In our study, the G-banding patterns completely coincided in the gonial and embryonic cells of all studied specimens. This means that all the karyotypes of spermatogonia and primary spermatocytes we studied belonged to animals of the same species—*L. saxatilis*.

### 4.2. Male Meiosis Is Non-Canonical in Littorina saxatilis

In this paper, we distinguish between non-canonical meiosis and atypical meiotic stages. Non-canonical meiosis exhibits stages that are not characteristic of its mainstream course [4]. Atypical stages of meiosis are mentioned in two cases: (i) when an atypical prolongation of a stage occurs and several substages can be clearly distinguished based on the morphological characteristics of the bivalents; (ii) when an exotic structure of bivalents is observed at a certain stage. Reconstruction of meiotic stages using nuclear spreads of *L. saxatilis* germ cells has revealed for the first time some non-canonical features of male meiosis I in this mollusk species (Figure 11). In meiosis II, no deviations from the mainstream course of meiosis were detected.

In the present study, we discovered some specific and unusual cytogenetic structures in the following stages of *L. saxatilis* male meiosis:

**bouquet-like stage and zygotene**—short chains of chromomeric pairs with cross-bridges between them instead of small fibers of compact chromatin, resolved by conventional microscopy at the sites of primary pairing of homologous chromosomes;

**early pachytene**—atypically shortened and low spiralized bivalents, in which most homologous chromomeres are joined by cross-bridges, whereas the canonical stage is characterized by highly coiled bivalents with few cross-bridges;

**late pachytene**—necklace-like bivalents with lack the asynaptic regions characteristic of late pachytene bivalents in mainstream meiosis;

**mid pachytene**—bivalents with intermediate morphology compared to the previous and subsequent stages;

non-canonical and atypical **diffuse stage**—bivalents with decompacted chromatin in the central part and short loops of diffuse chromatin at the periphery;

atypical **diplotene**—shortened necklace-like bivalents with lateral chromatin loops;

non-canonical **karyosome stage**, including distinct substages of pre-karyosome, karyosome and post-karyosome; bivalents are assembled together into a compact chromatin mass;

**early diakinesis**—shortened bivalents consisting of densely packed chromatin; a set of bivalents contains only one, rarely two, cruciform bivalents with a single chiasma;

**mid diakinesis**—some bivalents are elongated and acquire a distinct chromomeric pattern, others have a cruciform or ring-shaped appearance with one or two chiasmata;

**late diakinesis**—elongated bivalents exhibit individual G-banding patterns; seven types of chiasmata are recognizable; some bivalents demonstrate a delay in the appearance of chiasmata;

**early metaphase I**—all bivalents have individual G-banding patterns, with the exception of three short bivalents that are indistinguishable from each other; chiasmata appear in most bivalents; one bivalent sometimes exhibits delayed chiasmal expression or completely dissociates into homologous chromosomes; bivalents with terminal filiform chiasmata exist;

**late metaphase**—long bivalents are identified by their specific G-banding patterns;

**early anaphase**—some bivalents of the set contain terminal filiform chiasmata, others show a strong separation of sister chromatid tips.

The diagram (Figure 11) generally corresponds to the classical descriptions of mainstream meiosis [1,2,3,4], with the exception of three additional stages. The first one is the interlocking (interlock resolution) stage, which occurs in the interval between zygotene and pachytene and characterizes many organisms [41,42,43,44,45]. With the introduction of high-resolution microscopy methods, it was found that, in addition to interlock formation between bivalents, there is a coiling of the axial structures of homologous chromosomes associated with the appearance of cross-bridges between them [53]. Removal of interlocks and unwinding of homologues towards mid-pachytene are associated with topoisomerase II activity and long-range movements of chromosome arms, recorded using 4D microscopy at the stage of interlock resolution and early pachytene [43,54].

The structure of bivalents at the interlock resolution stage of *L. saxatilis* male meiosis resembles the structure of early pachytene bivalents during canonical male meiosis in maize [7]. At mid pachytene, bivalents have a classic necklace-like appearance with fused homologous chromomeres. A similar structure was also recorded on bivalent spreads in humans, animals and higher plants, which made it possible to create chromomeric maps of pachytene chromosomes and perform cytogenetic mapping of chromosomal rearrangements with DNA probes [55,56,57,58]. However, in supernumerary B chromosomes of maize, unlike the conventional A chromosomes of the set, fusion of homologous chromomeres is never observed [59].

### 4.3. Two Ways of Bivalent Formation: Pairing and Synapsis of Homologous Chromosomes

According to modern concepts, the formation of bivalent in zygotene occurs in two stages [42,44,60]. In the first stage, called co-alignment or homologous chromosome pairing, homologous chromosomes gradually align next to each other. In the second stage, called synapsis, the formation of the SC occurs between the regions of homologues oriented side by side. In canonical meiosis, the distance between pairs of homologues does not exceed 200 nm [44]. When using the maximum resolution of conventional microscopy, paired regions of this type are recorded as one-thread chromatin fibrils that are twice as thick as unpaired fragments of homologous chromosomes (Figure 10—canonical zygotene). In a number of plants and fungi, in particular in the homothallic ascomycete *Sordaria macrospora*—a widely used model for studying the early stages of meiosis—the co-alignment distance is 400 nm and can reach 600–800 nm in some mutants without affecting their viability [42,60]. Paired regions of zygotene homologous chromosomes, organized as in *Sordaria*, are visualized as binemic chromatin structures. Similar structures were also visible in *L. saxatilis* zygotene, both in DIC-live images and in fixed spreads of primary spermatocytes.

In zygotene—early pachytene, *L. saxatilis* chromosomes exhibited another non-canonical feature of co-alignment, characteristic of the zygotene of S. macrospora. These are chromatin cross-bridges connecting homologous regions of paired chromosomes. A specific feature of the zygotene co-alignment of homologous chromosomes in *L. saxatilis* is the appearance of small chromomere-like structures of condensed chromatin in the chromosome pairing zones. These beads correspond in size to the chromomeres of pachytene chromosomes. Thus, zygotene pairing of homologous chromosomes in *L. saxatilis* is associated with chromatin condensation and the formation of short chromomeric fragments of pachytene chromosomes (Figure 10—non-canonical zygotene and early pachytene).

### 4.4. Bouquet-Like Stages of Meiosis

In most organisms, both telomeric ends of all chromosomes of the set move, forming a bouquet [5,6,7,61,62,63,64]. This results in the formation of a cluster of telomeres and subtelomeres. In some model organisms, instead of telomeres, a cluster of centromeres (*Drosophila*) or pairing centers (*C. elegans*) is formed. In the latter case, the pairing centers are rather large segments of chromosomes that map to a specific position on only one of the two telomeric ends of each chromosome in the set [64].

The existence of a bouquet stage in meiosis of *Littorina* males remains an open question, since we did not observe classical bouquet patterns in the nuclei of *L. saxatilis* spermatocytes. It is possible that they do not form a complete bouquet, as in *C. elegans* oocytes [64,65]. Unfortunately, the use of FISH with the canonical telomeric repeat (TTAGGG)n seems unpromising, since in the karyotype of *L. saxatilis*, according to our preliminary data, interstitial sites of this repeat are found in most chromosomes of the set. Furthermore, primary spermatocytes that could be identified as being at the bouquet stage were found in rare cells even in multicellular spermatocysts, and, moreover, not in all animals studied. Further studies are needed, including immunolocalization of bouquet-associated proteins—SUN1, KASH5 or telomere-binding proteins.

The duration of the telomeric bouquet stage in male meiosis can vary considerably, from late leptotene—early zygotene to the end of early pachytene, at least in maize [41]. Typically, the breakdown of the chromosome bouquet occurs at the stage of interlock resolution [42,43,44,45]. The prolongation of the chromosome bouquet period in male meiosis correlates (in maize) with a high level of coiling of homologous chromosomes in early pachytene—at least 6–8 turns per bivalent—as well as with the existence of cross-bridges between homologous chromomeres near the sites of twisting of such chromosomes [41].

In oocytes of the nematode *C. elegans*, pairing centers do not form a cluster, although they do interact with each other to form small groups [65]. The authors of this work believe that the lack of clustering reflects centrosome inactivation and the resulting lack of a cytoplasmic focus of microtubules. This hypothesis is supported by data from the analysis of a spectrum of mutants in budding yeast, which revealed that pairing efficiency was better correlated with the rate of meiotic chromosome motion than the degree of telomere clustering [66]. There is an opinion that the meiotic bouquet is not necessary for either homologous pairing or synapsis, but accelerates and increases the efficiency of both processes [6].

### 4.5. Atypical Pachytene in Littorina saxatilis Male Meiosis

In *L. saxatilis* spermatocytes, in contrast to the canonical male meiosis [41], we observed a decrease in the level of homologous chromosome spiralization during the transition from the interlocking stage to early pachytene (Figure 10—early, mid and late pachytene). Cross-bridges between homologous chromomeres do not disappear. In mid pachytene, part of the bivalents and/or their large segments retains both pairs of homologous chromomeres and cross-bridges. The remaining bivalents and/or their segments showed fused homologous chromomeres.

It is known that the presence of cross-bridges prevents the synapsis of homologous chromosomes. For example, SCs found in bivalents at the early pachytene stage of male meiosis in maize are discontinuous [41]. According to the cited work, a continuous SC is formed in the mid-pachytene. Bivalents at this stage consist of fused chromomeres. Similarly, synapsis of homologous chromosomes in *Sordaria* meiosis occurs only after the removal of cross-bridges between the axial structures of paired homologous chromosomes [42]. As a result of asymmetric SC disassembly in late pachytene of canonical meiosis, areas of asynapsis of homologous chromosomes appear in bivalents, marking this stage [67]. There are no such areas in the late pachytene bivalents in the spermatocytes of *L. saxatilis*. The necklace-like bivalents exhibit only fused chromomeres.

### 4.6. G-Banding Patterns of Individual Pachytene Bivalents and High-Resolution Mitotic Chromosomes

Such comparisons have so far been performed between bivalents of mid- and late-pachytene spermatocytes/oocytes and early metaphase chromosomes of human somatic cells [68,69]. In these cases, the G-banding patterns of individual chromosomes and bivalents of the set generally coincide. However, differences have been noted between some chromomeres and G-bands occupying homologous positions on maps/idiograms of the individual chromosome. These differences concerned both the resolution of substructures in an individual G-band/chromomer and their relative sizes and intensity of staining with Giemsa dye. As in *L. saxatilis*, the resolution of individual human pachytene chromosomes—the number of chromomeres per bivalent/G-banded chromosome—was greatest for oocytes and approximately the same for spermatocytes and early metaphase G-banded chromosome. The resolution of mid/late pachytene spermatocyte bivalents of the mouse, domestic pig, Chinese and Turkish hamsters was significantly lower than the resolution of standard G-banded chromosome idiograms and the corresponding spread chromosomes [55,70,71,72].

Apparently, the situation described earlier for humans [70] and in the present work for *L. saxatilis* is a rare case of coincidence of both the resolution and patterns of G-bands and chromomeres in individual pachytene bivalents of spermatocytes and homologous early metaphase chromosomes of somatic cells, respectively. In addition, we have found that the chromomere/interchromomer pattern and the resolution of individual bivalents of *L. saxatilis* spermatocytes in early and late pachytene are identical.

### 4.7. Non-Canonical and Atypical Diffuse Stage in L. saxatilis Male Meiosis

During non-canonical diffuse stage of meiosis I, chromatin becomes relaxed, allowing the transition from pachytene to diplotene [25,26,27]. This stage, which some researchers consider to be the initial “diffuse” stage of diplotene, is characteristic of species with large genomes, in particular barley and wheat [27]. In cereals, the loosening of bivalent chromatin at this stage is associated with the selective removal of proteins of the central element of the SC and the formation of new “tinsel-like” structures, in which ASY1—one of the proteins of the axial component of the SC—apparently plays an important role [27].

The chromatin pattern in *L. saxatilis* bivalents at the diffuse stage of male meiosis is atypical. In the central part of one-thread bivalents, chromatin are loosened heterogeneously along the longitudinal axis of the bivalent, much weaker than at the periphery of the bivalent, where uniform short loops of diffuse chromatin are observed. The typical diffuse stage is characterized by the opposite picture, when the concentration of compact chromatin at the periphery of bivalents is noticeably higher than in their central parts.

### 4.8. Atypical Diplotene in Male but Not Female Meiosis of Littorina saxatilis: The Karyosome Stage

Our data also indicate that the non-canonical and/or atypical course of pachytene–early diakinesis period is characteristics exclusively of *L. saxatilis* males, whereas female meiosis is rather canonical in this species of mollusks (Figure 6).

We primarily found that male meiosis of *L. saxatilis* is characterized by a pronounced karyosome stage. The karyosome—a tangle of chromosomes assembled tightly into a compact mass—has been discovered first exactly in the spermatocyte nucleus, but more typically forms during oogenesis [22,23]. The data on karyosome formation in male meiosis are extremely scarce. In flies, the karyosome appears in the diplotene spermatocyte nucleus of *Drosophila virilis*, but not *D. melanogaster* [73]. However, in *D. virilis* spermatocytes, only part of the chromatin is compacted to form the karyosome, while the other part remains diffusely distributed throughout the nucleus.

In the oogenesis of most animals, if a karyosome is formed, individual chromosomes are often not distinguishable morphologically because they are tightly packed into a compact mass of chromatin [22]. This is reminiscent of the case of experimental depletion of condensin in mitotic cells [74], allowing to suppose that condensins may play a role in karyosome formation.

It is known that in the oogenesis of a number of invertebrate and vertebrate animals, the karyosome is formed independently of the existence of lampbrush chromosomes, but the lampbrush stage, if exist, precedes karyosome formation [22]. It is noteworthy that before the formation of the karyosome, diplotene bivalents of *L. saxatilis* spermatocytes have a fuzzy structure reminiscent of the lateral-loop structure of lampbrush chromosomes [16]. The formation of the karyosome, which follows the lampbrush stage, is first accompanied by constriction of lampbrush chromosomes’ lateral loops. However, the question of whether lampbrush chromosomes can actually be represented in the meiosis of males of any animal remains open, since the morphological similarities documented in many papers are only descriptive, especially with regard to the lateral “fuzz” of the chromosomes [16]. At the same time, the lampbrush-like loops are thought to be a conserved feature of the nucleus of primary spermatocytes, e.g., in many, if not all, Drosophilidae [75].

The conspicuous karyosome stage characterizes male, but not female, meiosis of the nematode *C. elegans* [24]. In spermatogenesis of this worm, the karyosome stage is rather prolonged, follows conventional diplotene stage, and precedes diakinesis. The assembly of *C. elegans* spermatocyte chromosomes into the karyosome initiates following disassembly of the SCs. However, the presence of core SC components is not a prerequisite for karyosome formation in *C. elegans* spermatocytes [24], as also shown for *Drosophila* oocytes [76].

An essential morphological difference between the *L. saxatilis* male karyosome and the karyosome in female meiosis of any organism [22,23] or between the male karyosome of *D. virilis* [73] and *C. elegans* [24] concerns a close association of the *L. saxatilis* male karyosome with the nuclear envelope of spermatocytes (Appendix A), which persists throughout the entire karyosome stage. Noticeably, Blackman—the discoverer of the karyosome—had also observed gradual accumulation the chromatin “*in one large, intensely staining body situated peripherally in close contact with the nuclear membrane*” in the spermatocyte nucleus of the giant centipede, *Scolopendra heros* [77].

### 4.9. Atypical Diakinesis in L. saxatilis Male Meosis: Inverse Order of Chromatin Compaction and Delayed Expression of Chiasmata

Conventional diakinesis is not usually divided into early and late stages. The gradual increase in the level of chromatin compaction of bivalents and a decrease in their length that occurs during diakinesis usually does not allow establishing clear criteria for identifying such stages in the bivalent spreads of individual diakinetic spermatocytes. In the spreads of primary spermatocytes of *L. saxatilis*, on the contrary, we were able to distinguish early, mid and late stages of diakinesis. Early diakinetic bivalents are greatly shortened, uniformly colored, have no internal split with the exception of the central regions of individual large bivalents. In addition, they have one, or rarely two, cruciform chiasmata in the shortest chromosomes of the set. In mid diakinesis, chromosomes represent elongated bivalents with or without chromomere structure. They also bear chiasmata of two to three variants. In late diakinesis, bivalents reach their maximum size, while all of them have a chromomeric structure and—with the exception of the 3–5 largest bivalents of the set—have an internal split along the entire length, with the exception of the chiasma region. The number of variants of chiasmata reaches eight—the maximum possible for *L. saxatilis* spermatocytes.

There are no cases described in the literature of preservation of the chromomeric structure of bivalents during diakinesis. Moreover, the chromomeric structure is not expressed even in diplotene spermatocytes. The only exception is human spermatocytes, for which chromomeric maps of diplotene bivalents have even been compiled [78].

Chiasmata are usually counted on the spreads of male meiocytes that were in metaphase I at the time of fixation. There are only a few exceptions to this rule. One of them is maize, for which an image classifier of chiasma-bearing bivalents has even been compiled, intended for quantitative accounting of chiasmata in diakinesis [39].

Interestingly, in *L. saxatilis* we found all the chiasmatic types of diakinetic bivalents found in maize, plus two types of bivalents not recorded in it. According to modern concepts, the shape of chiasma-bearing bivalents is determined, in addition to the number of chiasmata, by the asymmetrical nature of the disassembly of the SCs [79]. The shape of the bivalent can be completely different with the same number of chiasmata, depending on the position of the retained part of the SC. For example, with one chiasma per bivalent, it can be cruciform, rod-shaped, or H-shaped, depending on the position of the residual SC and chiasma on the chromosome or complete disassembly of the SC. According to the authors of the cited work, SC remnants are retained between the centromeric regions of homologous chromosomes until anaphase I of conventional meiosis, when they are finally removed. However, in the case of *L. saxatilis* spermatocytes this is not the case, at least for the meta- and submetacentric chromosomes of the set, in which these regions segregate in early diakinesis, while in the rest of bivalents homologous chromosomes can be in close association with each other until the end of diakinesis. Considering the high level of chromatin condensation of bivalents at the karyosome stage, the presence of residual SCs in diakinetic bivalents seems doubtful; therefore, the topological diversity of crossover bivalents in diakinesis should be explained by the involvement of factors other than SC components. Apparently, the only mechanism that can prevent premature segregation of homologous chromosomes in *L. saxatilis* male meiosis I is the formation of the so-called obligate chiasma [80].

### 4.10. Atypical Features of Metaphase and Anaphase I in Littorina saxatilis Male Meiosis

An atypical feature of metaphase I in *L. saxatilis* male meiosis is the chromomeric organization of bivalents, which makes it possible to identify the majority of bivalents using the Image Bank of individual chromosomes (Figure 6). In early metaphase I, it becomes possible to count the number of chiasmata with delayed segregation of homologous chromosomes, which masks the position of the chiasmata. The remaining bivalents contain exclusively poorly visible terminal chiasmata. In late metaphase I, the chromatin of the bivalents is compacted, and the bivalents largely lose their chromomeric organization. An atypical feature of anaphase I is the dissociation of sister chromatids, with the exception of their centromere regions. This is accompanied by the running away of their free tips in most homologous chromosomes of the set, which can indicate the possible rapid cleavage of cohesin bonds between the chromatids along the entire length of the chromosome, excluding pericentromeric regions resistant to separase [81].

## 5. Conclusions

The mechanisms responsible for the unusual course of male meiosis in *L. saxatilis*, including the possible role of cohesion and shelterin complexes, as well as SCs and/or the alternative homolog conjunction (AHC) complex in the over compaction of bivalents during late pachytene—early diakinesis, remain to be elucidated using ultrastructural, immunocytochemical, and molecular methods. It is possible that the AHC complex [81] or its functional analog may be a factor in the increased compaction of bivalents in prophase I of male meiosis in *L. saxatilis*, identified in the present study.

The marine gastropod *L. saxatilis* is a promising model object for evolutionary biology. Our work provides a cytological basis for further studies of conservation/variability and evolution of male meiosis. The described fact or scenario of unconventional male meiosis in *L. saxatilis* should be taken into account in future studies of Littorinidae. Along with *L. arcana* and *L. compressa*, *L. saxatilis* is a member of the sympatric cryptic species complex “saxatilis.” Analysis of meiotic features in other species of the “saxatilis” complex, as well as in other species of the genus *Littorina*, would be promising, and the mechanisms underlying reproductive and breeding strategies remain the focus of current research [32,82].

## Figures and Tables

**Figure 1 biology-14-01572-f001:**
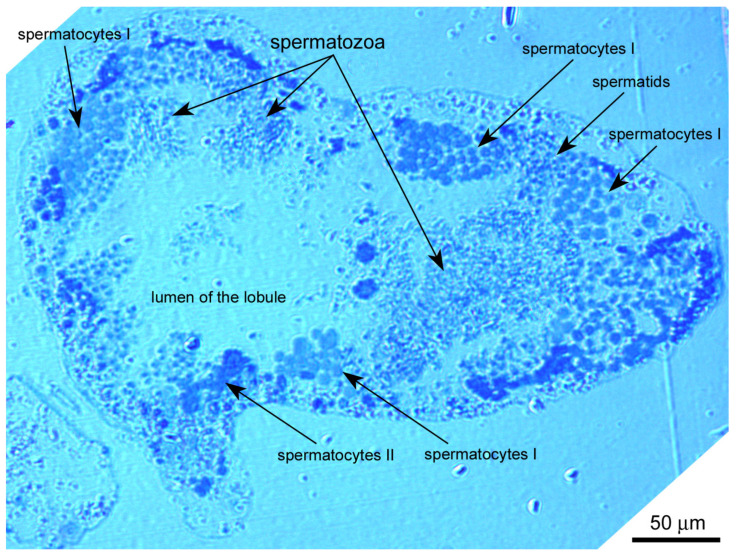
Semithin section of the *Littorina saxatilis* testis lobule, demonstrating male germline cells at different stages of spermatogeneis. Epon embedding, toluidine blue staining.

**Figure 2 biology-14-01572-f002:**
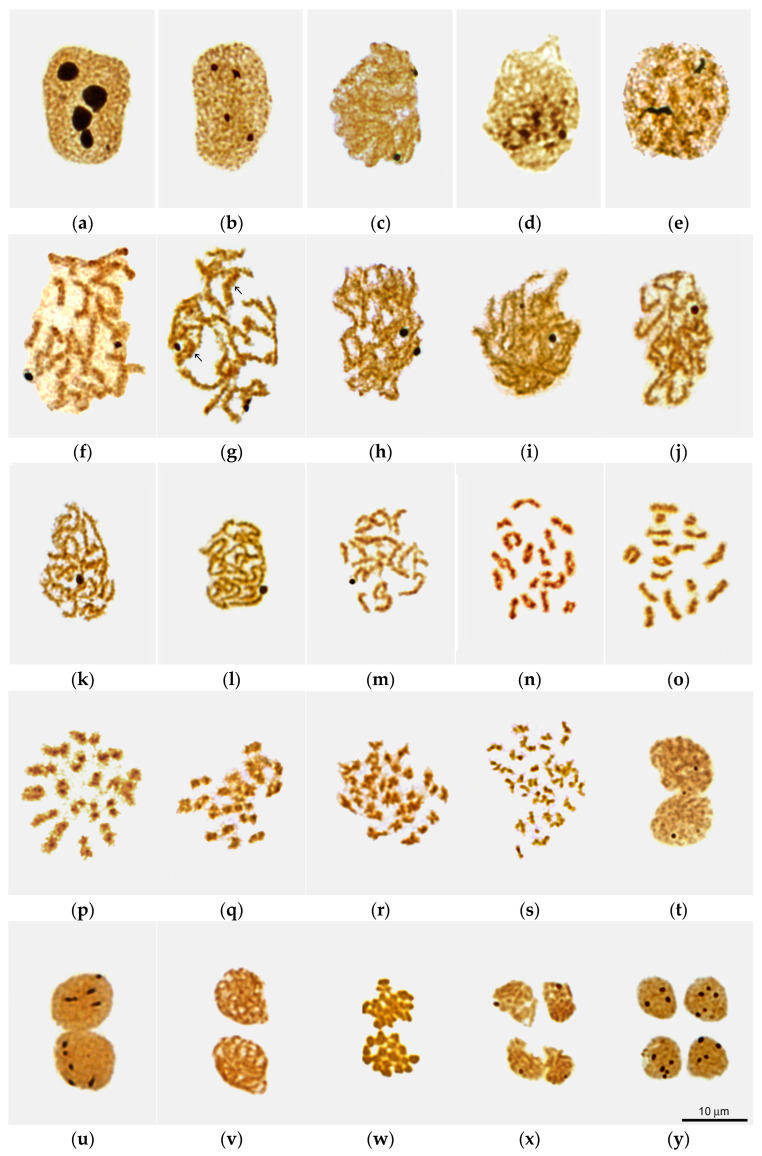
Sequential stages of *Littorina saxatilis* male meiosis as viewed in nuclear spreads after silver staining: (**a**) putative interphase spermatogonium; (**b**–**t**) primary spermatocytes during interphase (**b**), leptotene (**c**), bouquet-like stage (**d**), zygotene (**e**), early (**f**), mid (**g**) and late (**h**) pachytene, diffuse stage (**i**), atypical diplotene with compact fuzzy bivalents (**j**), pre-karyosome (**k**), mid karyosome (**l**) and post-karyosome (**m**) stages, early (**n**), mid (**o**) and late (**p**) diakinesis, early (**q**) and late (**r**) metaphase I, anaphase I (**s**), and telophase I (**m**); (**u**–**y**) secondary spermatocytes during interphase (**u**), prophase II (**v**), metaphase II (**w**), anaphase–telophase II transition (**x**), and late telophase II (**y**). Arrows in (**g**) point to banded bivalents.

**Figure 3 biology-14-01572-f003:**
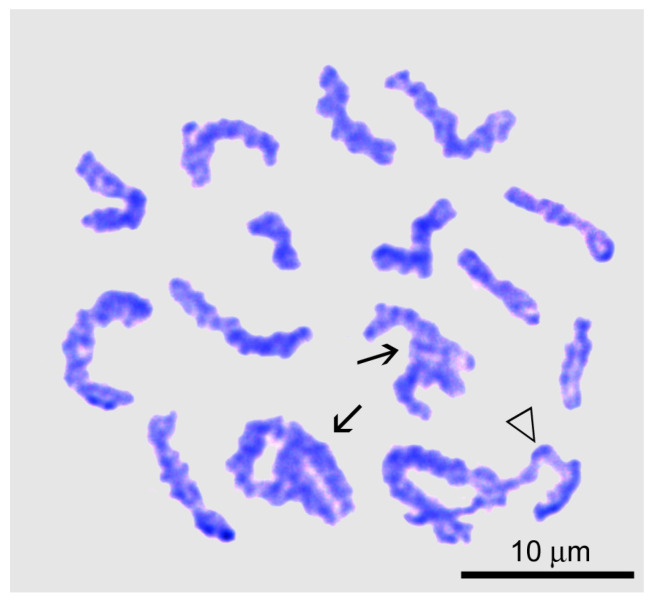
Chromosome spread of a primary spermatocyte at the interlock resolution stage of *Littorina saxatilis* male meiosis. Entangled bivalents are marked by arrows; the triangle indicates the bivalent that is essentially desynapted into homologous chromosomes during interlock resolution.

**Figure 4 biology-14-01572-f004:**
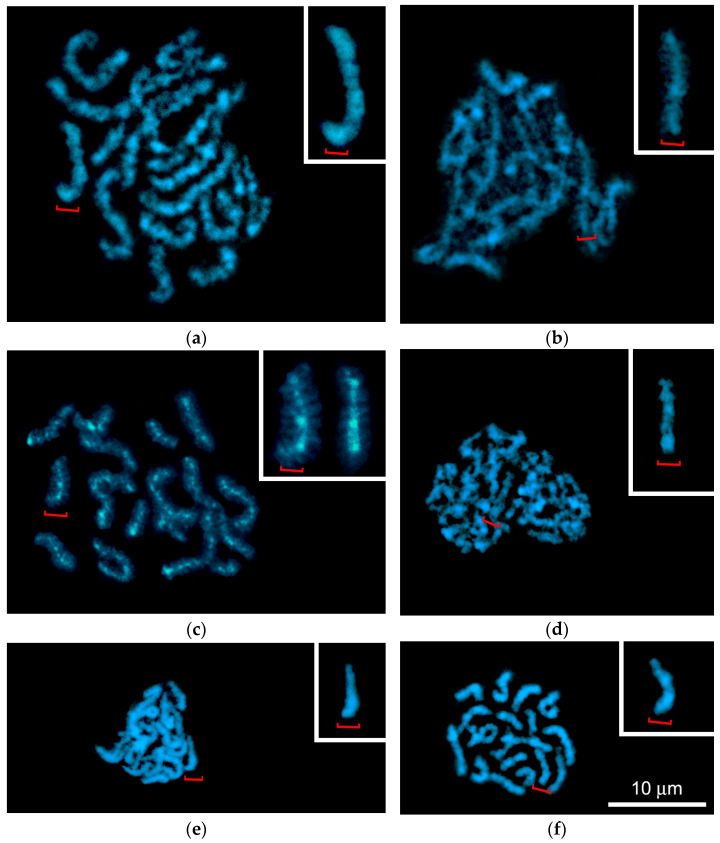
Chromosome spreads of *Littorina saxatilis* primary spermatocytes after DAPI staining, demonstrating bivalents during six sequential steps of meiosis: (**a**) late pachytene with banded bivalents; (**b**) diffuse stage; (**c**) atypical diplotene with compact bivalents exhibiting lateral chromatin loops; (**d**–**f**) karyosome stage, including pre-karyosome (**d**), complete karyosome (**e**) and post-karyosome (**f**) substages. The insets in (**a**–**f**) show enlarged images of medium-sized bivalents marked with red brackets. Inset in (**c**) additionally shows a bivalent homologous to the bivalent marked with a red bracket.

**Figure 5 biology-14-01572-f005:**
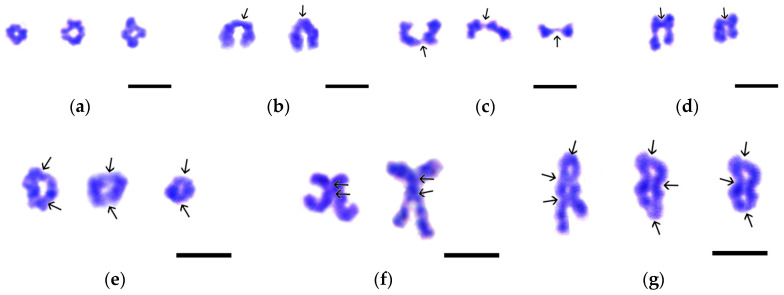
Variants of chiasma-bearing bivalents found in the nuclear spreads of *Littorina saxatilis* spermatocytes at the stages of diakinesis—metaphase I. (**a**) Cruciform bivalents with a single chiasma. (**b**) Typical rod-shaped bivalents with a single terminal chiasma. (**c**) Bivalents with a single filiform chiasma at the end of chromosomes. (**d**) Bivalents with a single interstitial chiasma. (**e**) Ring-shaped bivalents with two chiasmata, one at each end of the chromosome. (**f**) X-shaped bivalents with two closely spaced chiasmata. (**g**) Bivalents with three chiasmata: one terminal and two interstitial (left image) or one interstitial and two terminal (central and right images). Chiasmata are indicated by arrows. Chromatin was stained by Giemsa dye. Bars represent 5 μm.

**Figure 6 biology-14-01572-f006:**
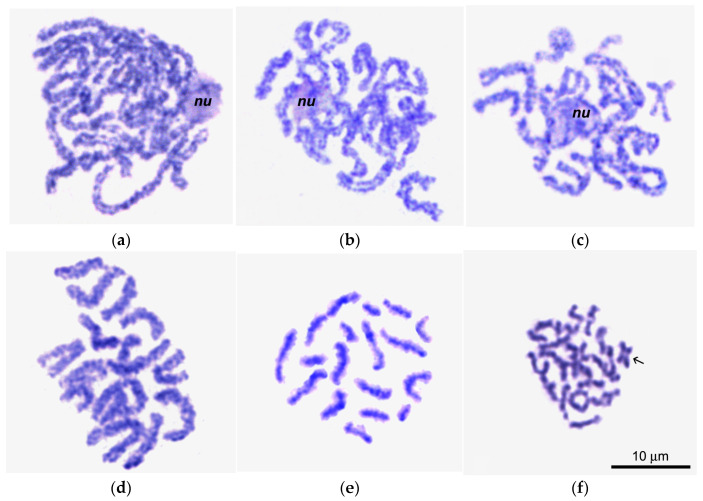
Giemsa-stained chromosome spreads of primary oocytes (**a**–**c**) and spermatocytes (**d**–**f**) in pachytene (**a**,**d**), diplotene (**b**,**e**), and early diakinesis (**c**,**f**); nu in (**a**–**c**), nucleolus; arrow in (**f**) points to a cruciform bivalent with a single chiasma.

**Figure 7 biology-14-01572-f007:**
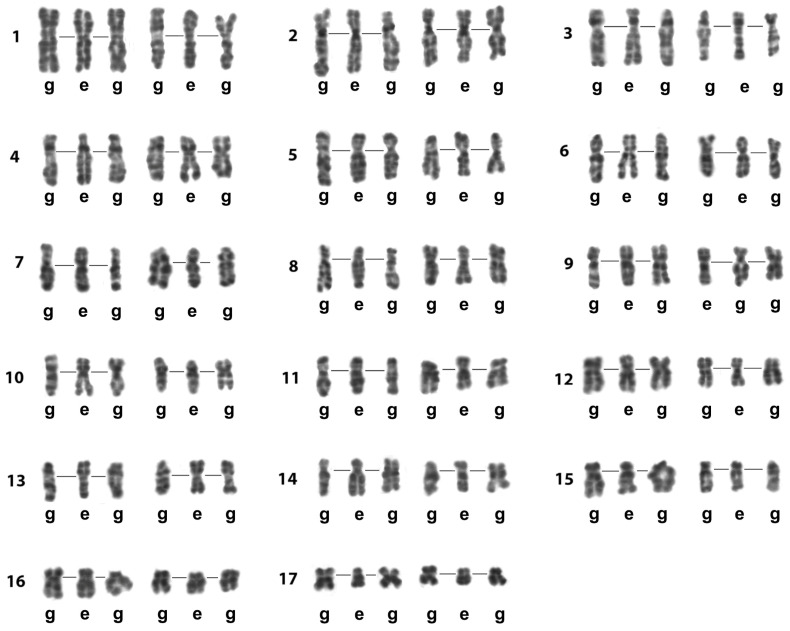
The Image Bank of *L. saxatilis* individual chromosomes based on their G-banding high-resolution patterns. Presented chromosomes are G-banded and computer straightened. Their original sources were the images of spread gonial (g) and early embryo (e) mitotic cells. Lines in the chromosome triplets join the centromere regions. Rows of individual chromosomes of *L. saxatilis* set (1–17) are numbered on the left.

**Figure 8 biology-14-01572-f008:**
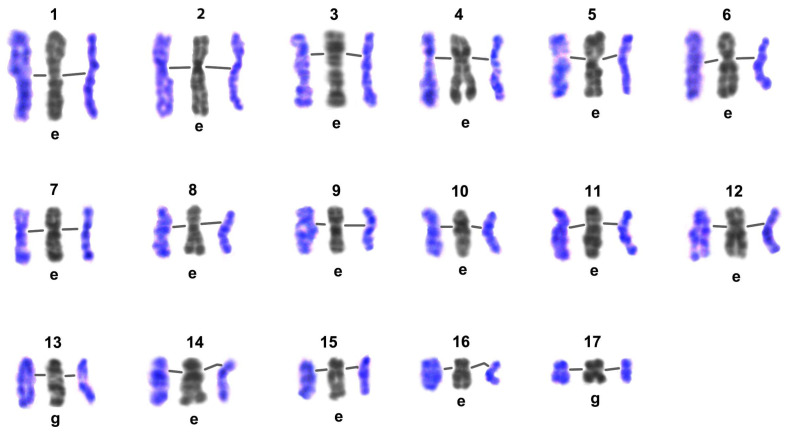
Comparison of G-band patterns of mitotic chromosomes and pachytene bivalents of *Littorina saxatilis* males. Individual chromosomes of the set are numbered (1–17); lines in the triplets of chromosome/bivalents join the centromeric regions; the left color image in each group represents the early pachytene bivalent, the right one represents the late pachytene bivalent, central gray images—straightened high-resolution mitotic chromosomes from early embryo (e) or gonial (g) cells—were taken from the Image Bank (Figure 7).

**Figure 9 biology-14-01572-f009:**
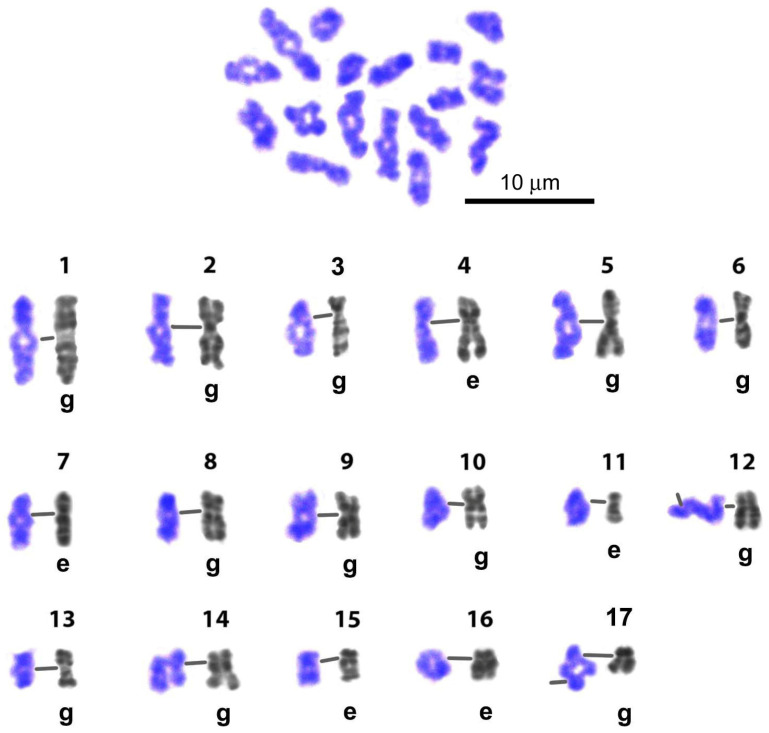
Representative karyogram of late diakinesis set of *L. saxatilis* G-banded spermatocyte bivalents. Gray numbered images (1–17)—straightened high-resolution individual mitotic chromosomes from early embryo (e) or gonial (g) cells—were taken from the Image Bank (Figure 6) and used for identification of individual bivalents. Line in the chromosome/bivalent pair joins the centromeric regions. Chiasmata expression in 13 bivalents is delayed.

**Figure 10 biology-14-01572-f010:**
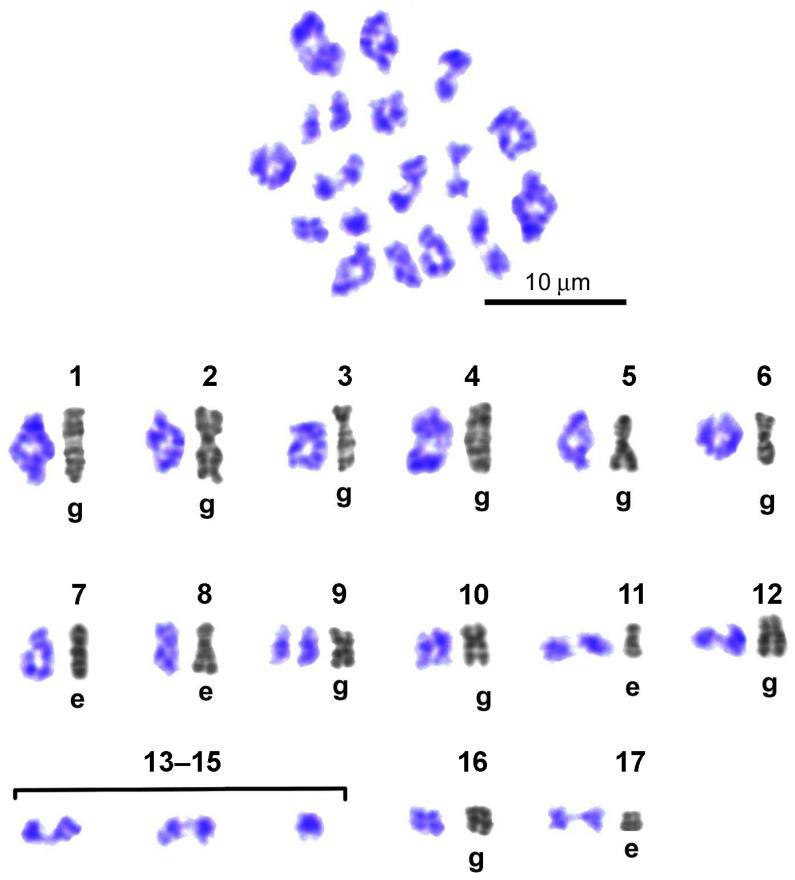
Representative karyogram of G-banded bivalents in *Littorina saxatilis* metaphase I set. Numbered gray images (1–17) represent high-resolution individual mitotic chromosomes of early embryo (e) or spermatogonial cells (g), obtained from the Image Bank (Figure 6) and used to identify individual bivalents. Bivalents ## 13–15 have indistinguishable G-band patterns. Bivalents ## 1–12, 16, and 17 have individual G-band patterns. Bivalent # 9 broke up into homologous chromosomes (univalents). Expression of chiasmata is delayed only in the case of bivalents # 8.

**Figure 11 biology-14-01572-f011:**
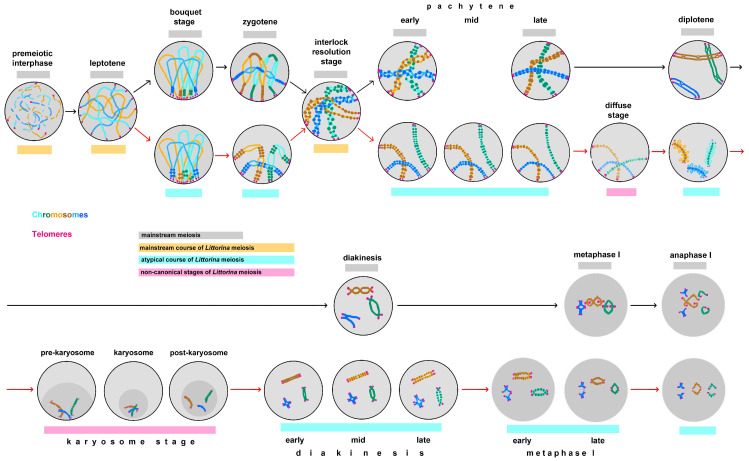
Male meiosis of the marine gastropod *Littorina saxatilis*, showing both mainstream and non-canonical stages. Gray lines highlight the stages characteristic of meiosis in general; yellow lines indicate the mainstream course of *L. saxatilis* meiosis—premeiotic interphase, leptotene and interlock resolution stage; cyan lines highlight atypical stages of *L. saxatilis* meiosis—bouquet stage, zygotene, diplotene, prolonged diakinesis—metaphase I and anaphase I; magenta lines indicate non-canonical stage of *L. saxatilis* meiosis—the diffuse stage and prolonged karyosome stage. Black arrows indicate the course of the mainstream male meiosis; red arrows show deviations from this mainstream in *L. saxatilis*.

## Data Availability

The data presented in this study are available on request from the corresponding authors.

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
