# Peer review of "Non-Canonical Male Meiosis in a Marine Gastropod, Littorina saxatilis"

_biology, 2025, doi:10.3390/biology14111572_

Round 1

Reviewer 1 Report

Comments and Suggestions for Authors

The paper presents a comprehensive cytological investigation of male meiosis in the marine snail Littorina saxatilis, revealing for the first time its non-canonical features. The authors used a combination of classical cytogenetic methods (Giemsa, DAPI, AgNOR staining), live imaging and high-resolution G-banding to provide a detailed description of the chromosome appearance at successive stages of male meiosis. They pay special attention to a description of its non-canonical features, such as diffuse and karyosome stages, delayed chiasma formation and atypical bivalent structures. These findings are documented with microscopy images.

The discussion considers the findings in the context of existing literature on canonical and non-canonical meiosis, discusses the evolutionary significance of observed features and suggests a model of chromosome pairing and segregation in this species.

Thus, the paper expands our knowledge on the variability of meiosis in animals and provides a framework for future studies in L. saxatilis, a prospective biological model. For these reasons, the paper deserves to be published in Biology.

However, I have to mention several issues of the paper, which could be improved.

1. The manuscript is far too long (37 pages!). The descriptions of stages and structural features of the chromosomes are too wordy and overloaded with technical terminology. The manuscript is very dense and at times repetitive. The results section is very long and contains extensive interpretations that might be better placed in the discussion. The paper would benefit from clearer, more concise language.

2. The study is predominantly descriptive cytology.It lacks molecular data (e.g., markers of double-strand breaks, proteins involved in meiosis) that could strengthen causal interpretations of the atypical stages or delayed chiasmata formation. The proposed Bivalent Compactization (BC) Complex model (Figure 12c) is intriguing but purely speculative based on cytological appearance.

There is no molecular data to support its existence. I would suggest framing the model more explicitly as a hypothesis to be tested in future work using modern methods of molecular cytogenetics, such as immunolocalization of known meiotic proteins (e.g., components of the synaptonemal complex - SYCP1, SYCP3; cohesins - Rec8; condensins) to see if their localization aligns with the observed structures (e.g., the cross-bridges, the BC complex hypothesis); electron microscopy to confirm the presence or absence of the synaptonemal complex during the atypical pachytene and the nature of the cross-bridges; FISH with telomeric repeats to clarify chromosome organization in the bouquet-like stage and the karyosome.

The dismissal of telomere FISH as "unpromising" due to interstitial signals is not fully convincing, as careful analysis can often distinguish telomeric clusters. A more robust attempt to characterize this stage is needed. immunolocalization of bouquet-associated proteins (e.g., SUN1, KASH5, or telomere-binding proteins) could be used to definitively determine if telomeres are clustered, even if FISH is challenging.

3. Although qualitative observations are strong, the paper could benefit from more quantitative data on frequencies of atypical stages, chiasmata counts, or statistical comparisons of male vs female meiosis to support conclusions more rigorously.

4. While evolutionary significance is mentioned, more explicit connections of these meiotic peculiarities to reproductive isolation, speciation, or life-history traits of L. saxatilis could frame the importance of the findings for a broader audience.

Minor Technical Point:

Simple summary is not simple at all. It is littered with obscure terminology.

Abstract is written mainly in passive voice and overloaded with unnecessary details.

All figures (especially the model in Figure 11/12) should be referenced in a logical order within the text. The current flow can be hard to follow.

The resolution of some figures (as they appear in the provided text) is low, making it difficult to appreciate the detailed chromomere patterns that are central to the paper. Please ensure that all figures, especially the Image Bank (Figure 7) and the karyograms (Figures 9, 10), are of the highest possible resolution in the final publication. Insets with higher magnification are helpful but should be clearer.

Comments on the Quality of English Language

The quality of the English language is quite poor, characterized by excessively long and cumbersome sentences, an overreliance on passive voice, and inconsistent use of articles. The paper definitely requires a language revision

Author Response

Thank you for critical reading of our MS and valuable comments. Below are our responses to the criticisms.

Comments 1. The manuscript is far too long (37 pages!). The descriptions of stages and structural features of the chromosomes are too wordy and overloaded with technical terminology. The manuscript is very dense and at times repetitive. The results section is very long and contains extensive interpretations that might be better placed in the discussion. The paper would benefit from clearer, more concise language.

Response 1. The manuscript has been shortened by more than 6 pages in accordance with the recommendations of the reviewer.

Comments 2. The study is predominantly descriptive cytology.It lacks molecular data (e.g., markers of double-strand breaks, proteins involved in meiosis) that could strengthen causal interpretations of the atypical stages or delayed chiasmata formation. The proposed Bivalent Compactization (BC) Complex model (Figure 12c) is intriguing but purely speculative based on cytological appearance.

There is no molecular data to support its existence. I would suggest framing the model more explicitly as a hypothesis to be tested in future work using modern methods of molecular cytogenetics, such as immunolocalization of known meiotic proteins (e.g., components of the synaptonemal complex - SYCP1, SYCP3; cohesins - Rec8; condensins) to see if their localization aligns with the observed structures (e.g., the cross-bridges, the BC complex hypothesis); electron microscopy to confirm the presence or absence of the synaptonemal complex during the atypical pachytene and the nature of the cross-bridges; FISH with telomeric repeats to clarify chromosome organization in the bouquet-like stage and the karyosome.

The dismissal of telomere FISH as "unpromising" due to interstitial signals is not fully convincing, as careful analysis can often distinguish telomeric clusters. A more robust attempt to characterize this stage is needed. immunolocalization of bouquet-associated proteins (e.g., SUN1, KASH5, or telomere-binding proteins) could be used to definitively determine if telomeres are clustered, even if FISH is challenging.

Response 2. In accordance with the need to accurately link the immunolocalization signals of meiotic proteins and FISH probes to the stages of meiosis and spermatogenesis, which we described earlier (Demin et al., 2019 Spermatogenesis and lobular cyst type of testes organization in marine gastropod Littorina saxatilis (Olivi 1792). Cell Tissue Res. 376(3):457-470), we deliberately limited ourselves to a thorough morphological description of male meiosis using a significant amount of data from lifetime observations. This was done in order not to burden the molecular studies planned by us and other researchers with a description of the stages of germ cell differentiation in male meiosis. Unfortunately, however, L. saxatilis cannot be stimulated to reproduce in the laboratory, so we have to work with field collection material during the mass reproduction of this mollusk. The nearest possible date for such fees is May 2026. It should also be noted that the Norwegian population we studied is unavailable to us for sanctions reasons, and only the populations of the Barents and White Seas remain at our disposal.

Comments 3. Although qualitative observations are strong, the paper could benefit from more quantitative data on frequencies of atypical stages, chiasmata counts, or statistical comparisons of male vs female meiosis to support conclusions more rigorously.

Response 3. Correct quantitative data cannot be obtained without a special methodological study, which takes into account the interindividual variability in the ratio of different stages of meiosis depending on the size of the gonads (age of the individual), as well as the high variability in the number of germ-line cells in individual cysts (the number of primary (secondary) spermatocytes per cyst within the gonad varies from 1 (4) up to 16 (64) and even up to 32 (128) in some animals). We do not have enough data to quantify the data of female meiosis, which is due to the fact that female meiosis is completed during the period of mass reproduction, so it is possible to find meiotic cells only in extremely rare cases. Nevertheless, we have included illustrations of pachytene, diplotene, and the diakinesis of female meiosis to stimulate future researchers to study it, since its course is radically different from that of male mitosis. We consider the current results an important discovery. Naturally, additional statistical confirmation is needed, as well as the use of molecular data for more rigorous proof. We view these steps as the next stage in the study of this fascinating natural phenomenon.

Comments 4. While evolutionary significance is mentioned, more explicit connections of these meiotic peculiarities to reproductive isolation, speciation, or life-history traits of L. saxatilis could frame the importance of the findings for a broader audience.

Response 4. Littorina saxatilis, along with L. arcana and L. compressa, is a member of the sympatric cryptic species complex "saxatilis." Members of these species lack precopulatory isolation. Reproductive isolation (also likely incomplete) is maintained by postzygotic mechanisms. It can be hypothesized that the discovered meiotic features accompany the evolutionary development of reproductive isolation within the species complex. From this perspective, analysis of meiotic features in other species of the "saxatilis" complex, as well as in other species of the genus Littorina, would be promising. We add this issue in the Conclusion section.

Minor Technical Point:

Simple summary is not simple at all. It is littered with obscure terminology.

Response. It has been shortened in accordance with the recommendations of the reviewer.

Abstract is written mainly in passive voice and overloaded with unnecessary details.

All figures (especially the model in Figure 11/12) should be referenced in a logical order within the text. The current flow can be hard to follow.

Response. Fig 12 deleted from the text when it was shortened.

The resolution of some figures (as they appear in the provided text) is low, making it difficult to appreciate the detailed chromomere patterns that are central to the paper. Please ensure that all figures, especially the Image Bank (Figure 7) and the karyograms (Figures 9, 10), are of the highest possible resolution in the final publication. Insets with higher magnification are helpful but should be clearer.

Response. Apparently, your version of the manuscript contains mid or even high packaged versions of the images. The images we have obtained have an unusually high resolution for invertebrate chromosomes. The Figures presented in the editorial version of the manuscript can be enlarged 10 times without noticeable pixilation.

Reviewer 2 Report

Comments and Suggestions for Authors

The work is based on a very interesting idea—studying meiosis in a non-model animal species belonging to a group in which meiosis has not been investigated at all. However, the way the material is presented and the quality of the figures do not meet the standards of the journal. I would suggest either publishing a simplified version in a cytogenetic journal, with a clear description of all observed phenomena (especially the “atypical stages”), or raising the level of the study by including an analysis of synaptonemal complexes, although I understand that this would require the production of customized antibodies.

There are many non-canonical terms that need clarification or definition, including “chiasmata expression”, “single-thread form of bivalents”,“double-thread form of bivalents”, “a narrow region of the nucleus”, ‘interchromomeres” — please, explain the terms or provide a reference 

Another undefined term is “the Image Bank”.

Figure 1. Please explain how spermatocytes I and spermatocytes II were distinguished. This is not clear from the figure.

Figure 2. Many stages (for example, j to m) appear very similar. The authors must provide evidence that these are not artifacts of differential chromosome spreading.

Line 188. “The nuclei of interphase spermatogonia were filled with diffuse chromatin, randomly distributed in the nuclear volume.” I believe that FISH with chromosome-specific paints would reveal chromosome territories. Please provide additional evidence to support the hypothesis of random chromatin distribution.

Line 196. “In contrast to spermatogonia, interphase primary spermatocytes could be frequently found in the testes of all specimens studied.” Does this mean that spermatogonia were absent in many individuals? Please discuss possible reasons for this observation.

Line 200. “The stages of leptotene and zygotene morphologically can be referred to as mainstream stages of L. saxatilis male meiosis.” On what basis are only these two stages considered “mainstream”? Are other stages considered atypical or unconventional?

Line 202. “single-thread chromosomes” — Do you mean single-chromatid chromosomes? Please clarify.

Line 210. “However, clusters of condensed chromatin, apparently reflecting the onset of synapsis, are visible.” The evidence does not convincingly support that this stage represents a bouquet.

Line 216. “These fragments are separated from each other, and one-thread fragments of unassociated homologs were observed.” Please use arrows in the figure to indicate univalent and bivalent parts.

Line 232. “At this stage of meiosis, free bivalents have a canonical pachytene organization—chromomeric structure and strong twisting of the axial structures of homologous chromosomes (2–6 turns per bivalent).” Please indicate these twists in the figure, as they are not visible.

Line 241. “Depending on the state of the chromosomal apparatus—namely, the degree of expression of a two-thread structure along the bivalents—the nuclei of spermatocytes that are at the prolonged pachytene stage can be divided into three categories: early, mid, and late.” The differences are not apparent in the provided images. Please use arrows to demonstrate the “degree of expression of the two-thread structure.”

Line 248. “The nuclei of late pachytene spermatocytes predominantly exhibited one-thread…” Could this be a spreading artifact?

Figure 4. “Inset in (c) additionally shows a bivalent homologous to the bivalent marked with a red bracket.” What exactly is marked in red? The figure is unclear. There appears to be little difference between the depicted stages—could these variations result from preparation artifacts?

Line 516. “This is consistent with recent data on the nonchromosomal type of sex determination in nine periwinkle species.” I strongly disagree with this statement, as there are numerous cases of homomorphic sex chromosomes that cannot be distinguished by cytogenetic methods.

Line 523. “who suggest that an XY sex determination system is not the case for periwinkles, since targeted re-sequencing did not reveal any regions with low heterozygosity in L. saxatilis males.” I disagree, as there are many XY species that do not show low heterozygosity on the Y chromosome.

Comments on the Quality of English Language

There are many non-canonical terms that need clarification or definition

Author Response

Thank you for critical reading of our MS and valuable comments. Below are our responses to the criticisms.

Comments 1. The work is based on a very interesting idea—studying meiosis in a non-model animal species belonging to a group in which meiosis has not been investigated at all. However, the way the material is presented and the quality of the figures do not meet the standards of the journal. I would suggest either publishing a simplified version in a cytogenetic journal, with a clear description of all observed phenomena (especially the “atypical stages”), or raising the level of the study by including an analysis of synaptonemal complexes, although I understand that this would require the production of customized antibodies.

Response1. Apparently, your version of the manuscript contains mid or even high packaged versions of the Figs. They are too large to be sent by usual e-mail browser, for example, the Fig 2 tiff file has a size of 6.44 Mb. In real size, they are twice as large as in the manuscript. Chromosome Images we have obtained have an extremely high resolution for invertebrate. They can only be obtained using the rarely used high pressure squashing technique (very slow pressure increase on the slide up to the level of 150-250 kg per square centimeter). The reconstruction of meiosis on spreads was carried out in accordance with the data of lifetime observations using DIC microscopy at the limit of its resolution. To discriminate between the stages of meiosis, only chromatin structures and their patterns recorded in intact nuclei of microsurgically isolated spermatocyte cysts were taken into account to avoid describing artifacts (See technical details in our previous papers: Demin et al., 2019 Spermatogenesis and lobular cyst type of testes organization in marine gastropod Littorina saxatilis (Olivi 1792). Cell Tissue Res. 376(3):457-470; Demin et al., 2020. New Data on Spermatogenic Cyst Formation and Cellular Composition of the Testis in a Marine Gastropod, Littorina saxatilisInt. J. Mol. Sci.21, 3792). Due to our oversight, during the reduction of the volume of the manuscript, representative lifetime DIC images of optical equatorial sections of spermatocytes cell nuclei of atypical and non-canonical stages, with the exception of the karyosphere stage, were inadvertently deleted. This naturally led to numerous questions from the reviewer. This error has been fixed in the edited version of manuscript (See Suppl Figure S2).

We also note that the manuscript has been shortened by more than 6 pages in accordance with the recommendations of the reviewer.

Comments 2. There are many non-canonical terms that need clarification or definition, including “chiasmata expression”, “single-thread form of bivalents”,“double-thread form of bivalents”, “a narrow region of the nucleus”, ‘interchromomeres” — please, explain the terms or provide a reference

Response 2. Outdated cytogenetic terms “single-thread form of bivalents”,“double-thread form of bivalents”, and also “a narrow region of the nucleus” removed from the manuscript. “Chiasmata expression" is not a term but two words from the newly described phenomenon “delayed chiasmata expression in diakinesis of L. saxatilis”. For term "interchromomer” a reference is provided.

Comments 3. Another undefined term is “the Image Bank”.

Response 3. “The Image Bank” it's short for “Image Bank of L. saxatilis individual chromosomes based on their G-banding high resolution patterns” that used to reduce the size of the text.

Comments 4. Figure 1. Please explain how spermatocytes I and spermatocytes II were distinguished. This is not clear from the figure.

Response 4. Spermatocytes II spreads differ from spermatocytes I ones in small size, AgNORs pattern, absence of bivalents, and clumpy mitotic figures in metaphase II and telophase II. We have previously published lifetime images of cysts with secondary and primary spermatocytes regarding the relative size and сlumpiness of the chromosome figures (See papers cited above).

Comments 5. Figure 2. Many stages (for example, j to m) appear very similar. The authors must provide evidence that these are not artifacts of differential chromosome spreading.

Response 5. See Response 1 and representative lifetime DIC images of optical equatorial sections of spermatocytes cell nuclei of atypical and non-canonical stages (new version of Suppl Figure S2).

Comments 6. Line 188. “The nuclei of interphase spermatogonia were filled with diffuse chromatin, randomly distributed in the nuclear volume.” I believe that FISH with chromosome-specific paints would reveal chromosome territories. Please provide additional evidence to support the hypothesis of random chromatin distribution.

Response 6. This is an unfortunate formulation. This error has been fixed in the edited version of manuscript.

Comments 7. Line 196. “In contrast to spermatogonia, interphase primary spermatocytes could be frequently found in the testes of all specimens studied.” Does this mean that spermatogonia were absent in many individuals? Please discuss possible reasons for this observation.

Response 7. The phenomenon of depletion of the pool of deferentiating spermatogonies during the reproduction season is described and discussed in our papers cited above. The renewal of the pool of spermatogonia in the next breeding season occurs due to dormant spermatogonia and small clusters of germ-line and cyst cell stem cells scattered along the lobules of the testis.

Comments 8. Line 200. “The stages of leptotene and zygotene morphologically can be referred to as mainstream stages of L. saxatilis male meiosis.” On what basis are only these two stages considered “mainstream”? Are other stages considered atypical or unconventional?

Response 8. These stages do not have clear morphological markers that allow us to assert their atypicity, unlike the subsequent stages of meiosis I. However, the structure of primary asynapsis zones in late zygote is closer to the non-canonical type of meiosis in Sordaria. It is possible that the immunolocalization of the corresponding meiotic proteins will allow these stages to be attributed to the Sordaria type.

Comments 9. Line 202. “single-thread chromosomes” — Do you mean single-chromatid chromosomes? Please clarify.

Response 9. This outdated term has been removed from the text. We had a necklace-like bivalent with fused homologous chromomerеs without a visually discernible split between the homologues

Comments 10. Line 210. “However, clusters of condensed chromatin, apparently reflecting the onset of synapsis, are visible.” The evidence does not convincingly support that this stage represents a bouquet.

Response 10. We agree with you, but we have not found a single cell with the structure of a bouquet of chromosomes among several thousand living primary spermatocytes in the testes of seven males and tens of thousands of spreads of fixed primary spermatocytes of several dozen males. The presented image is the closest approximation to the indicated stage.

Comments 11. Line 216. “These fragments are separated from each other, and one-thread fragments of unassociated homologs were observed.” Please use arrows in the figure to indicate univalent and bivalent parts.

Response 11. This fragment was removed from the manuscript when it was abridged.

Comments 12. Line 232. “At this stage of meiosis, free bivalents have a canonical pachytene organization—chromomeric structure and strong twisting of the axial structures of homologous chromosomes (2–6 turns per bivalent).” Please indicate these twists in the figure, as they are not visible.

Response 12. This fragment was removed from the manuscript when it was abridged.

Comments 13. Line 241. “Depending on the state of the chromosomal apparatus—namely, the degree of expression of a two-thread structure along the bivalents—the nuclei of spermatocytes that are at the prolonged pachytene stage can be divided into three categories: early, mid, and late.” The differences are not apparent in the provided images. Please use arrows to demonstrate the “degree of expression of the two-thread structure.”

Response 13. This fragment was removed from the manuscript when it was abridged.

Comments 14. Line 248. “The nuclei of late pachytene spermatocytes predominantly exhibited one-thread…” Could this be a spreading artifact?

Response 14. No. In the lifetime images of the nuclei of this stage, the bivalents have a similar structure (See renovated Suppl Figure S2, C). We had a necklace-like bivalent with fused homologous chromomerеs without a visually discernible split between the homologues.

Comments 15. Figure 4. “Inset in (c) additionally shows a bivalent homologous to the bivalent marked with a red bracket.” What exactly is marked in red? The figure is unclear. There appears to be little difference between the depicted stages—could these variations result from preparation artifacts?

Response 15. Bivalent marked with a red bracket belongs to the spread. The unmarked enlarged bivalent in the inset is taken from the spread of another cell in order to show that bivalents with  the similar chromeric pattern occur in different cells.

Comments 16. Line 516. “This is consistent with recent data on the nonchromosomal type of sex determination in nine periwinkle species.” I strongly disagree with this statement, as there are numerous cases of homomorphic sex chromosomes that cannot be distinguished by cytogenetic methods.

Response 16. We completely agree with you. Your ideas are in good agreement with the hypothesis of the origin of the Y chromosome from the X chromosome. This fragment was removed from the manuscript.

Comments 17. Line 523. “who suggest that an XY sex determination system is not the case for periwinkles, since targeted re-sequencing did not reveal any regions with low heterozygosity in L. saxatilis males.” I disagree, as there are many XY species that do not show low heterozygosity on the Y chromosome.

Response 17. We completely agree with you. This fragment was removed from the manuscript.

Reviewer 3 Report

Comments and Suggestions for Authors

The manuscript by Demin et al. shows a detailed cytogenetic analysis of male meiosis in L. saxatilis. The authors identified several specific and unusual features: the presence of a diffuse stage, a prolonged, multi-step karyosome stage, and delayed chiasmata expression. The authors used a combination of AgNOR staining, Giemsa staining, DAPI, and live imaging to observe the Sequential Stages of L. saxatilis Male Meiosis, and provided detailed description of these stages. In addition, the authors established the Image Bank of individual mitotic chromosomes of L. saxatilis with high-resolution G-banding which provide a valuable source for the L. saxatilis studies. These findings are novel and provide valuable insights to the understanding of meiotic diversity. The findings will be of interest to cytogeneticists and evolutionary biologists studying meiosis. However, the works are highly descriptive, the proposed model to explain homologous chromosome retention remains speculative, as it is not yet supported by any experimental evidence. It would be appreciated to either remove this statement from the abstract or reframe it more cautiously. Moreover, the authors described several intriguing cytological phenomena. Discussing the potential biological significance of these features would strengthen the manuscript's impact. Overall, I recommend acceptance of this manuscript with minor changes.

Main comments:

    The study identifies several non-canonical features of male meiosis. To further deepen the discussion, it would be insightful to explore the potential biological implications of these features. For instance, could these features be adaptations linked to the species' specific reproductive strategy?
    The authors mentioned the changes in nucleoli number during the meiosis; it would be valuable to place these observations in a comparative context by discussing whether the changing pattern is conserved in other known species. If not, the biological significance of these specific nucleolar dynamics should be discussed.

Minor comments:

    There is a repeat in the following part: “We further compared the structure of mitotic chromosomes of gonial and embryonic 428 cells, as well as pachytene bivalents in male meiosis of L. saxatilis, basing on their 429 G-banding patterns. (Figure 8).”

Author Response

Thank you for critical reading of our MS and valuable comments. Below are our responses to the criticisms.

Comments 1. The manuscript by Demin et al. shows a detailed cytogenetic analysis of male meiosis in L. saxatilis. The authors identified several specific and unusual features: the presence of a diffuse stage, a prolonged, multi-step karyosome stage, and delayed chiasmata expression. The authors used a combination of AgNOR staining, Giemsa staining, DAPI, and live imaging to observe the Sequential Stages of L. saxatilis Male Meiosis, and provided detailed description of these stages. In addition, the authors established the Image Bank of individual mitotic chromosomes of L. saxatilis with high-resolution G-banding which provide a valuable source for the L. saxatilis studies. These findings are novel and provide valuable insights to the understanding of meiotic diversity. The findings will be of interest to cytogeneticists and evolutionary biologists studying meiosis. However, the works are highly descriptive, the proposed model to explain homologous chromosome retention remains speculative, as it is not yet supported by any experimental evidence. It would be appreciated to either remove this statement from the abstract or reframe it more cautiously. Moreover, the authors described several intriguing cytological phenomena. Discussing the potential biological significance of these features would strengthen the manuscript's impact. Overall, I recommend acceptance of this manuscript with minor changes.

Response 1. Proposed model to explain homologous chromosome retention was removed from abstract and main text

Main comments:

Comments 2.    The study identifies several non-canonical features of male meiosis. To further deepen the discussion, it would be insightful to explore the potential biological implications of these features. For instance, could these features be adaptations linked to the species' specific reproductive strategy?

Response 2. The only thing we can add, without going into pure speculation, is that the described type of male meiosis provides more opportunities for its delay at non-canonical stages in the case of recurrent frosts or saline desalination of seawater during the breeding season.

Comments 3.    The authors mentioned the changes in nucleoli number during the meiosis; it would be valuable to place these observations in a comparative context by discussing whether the changing pattern is conserved in other known species. If not, the biological significance of these specific nucleolar dynamics should be discussed.

Response 3. We propose to do this in a separate paper with a comparative description of meiosis in closely related Littorina species.

Minor comments:

Comments 4.    There is a repeat in the following part: “We further compared the structure of mitotic chromosomes of gonial and embryonic 428 cells, as well as pachytene bivalents in male meiosis of L. saxatilis, basing on their 429 G-banding patterns. (Figure 8).”

Response 4. Error is fixed

Round 2

Reviewer 1 Report

Comments and Suggestions for Authors

The authors have done a good job addressing most of my comments, particularly in shortening the manuscript, improving the flow, and strengthening the evolutionary significance. However, the central concern regarding the speculative nature of the "Bivalent Compactization Complex" model remains largely unaddressed. My request was to reframe the model as a hypothesis, not necessarily to conduct the experiments immediately. While the authors’ logistical constraints are understood, they do not eliminate the need for scientific caution. As currently presented, the model reads more like a conclusion than a hypothesis.

I therefore recommend that the authors explicitly reframe the BC Complex model throughout the manuscript. The text should clearly state that this is a hypothetical model proposed to explain the cytological observations, and that its validation requires future molecular work (e.g., immunolocalization and EM studies I previously suggested). Without this crucial change in tone and framing, the manuscript overstates its findings.

Once this key issue is addressed, I believe the manuscript will be suitable for publication.

Comments on the Quality of English Language

The quality of the English is still quite poor, marked by excessively long and cumbersome sentences, overuse of the passive voice, and inconsistent use of articles. The paper definitely requires thorough language revision.

Author Response

Thank you again for your critical reading of our manuscript, valuable comments, and suggestions. Our response to your criticism is presented below.

Comment 1. The authors have done a good job addressing most of my comments, particularly in shortening the manuscript, improving the flow, and strengthening the evolutionary significance. However, the central concern regarding the speculative nature of the "Bivalent Compactization Complex" model remains largely unaddressed. My request was to reframe the model as a hypothesis, not necessarily to conduct the experiments immediately. While the authors’ logistical constraints are understood, they do not eliminate the need for scientific caution. As currently presented, the model reads more like a conclusion than a hypothesis.

I therefore recommend that the authors explicitly reframe the BC Complex model throughout the manuscript. The text should clearly state that this is a hypothetical model proposed to explain the cytological observations, and that its validation requires future molecular work (e.g., immunolocalization and EM studies I previously suggested). Without this crucial change in tone and framing, the manuscript overstates its findings.

Once this key issue is addressed, I believe the manuscript will be suitable for publication.

Response 1. The previous hypothetical ВС model has been removed from the text, as has the corresponding figure. Based on the cytological data presented in our MS, we can currently only point to studies that need to be considered or implemented to develop such a model. First of all, it is necessary to consider the existence of two ways of holding homologous chromosomes together in pachytene, metaphase I of male meiosis [ref. 82]. In the Conclusion section, we speculate that one of the factors for increased compaction of bivalents in prophase I of male meiosis in L. saxatilis could be the alternative homolog conjunction (AHC) complex, as occurs in achiasmatic meiosis of Drosophila [82], or its functional analogue.

Reviewer 2 Report

Comments and Suggestions for Authors

Line 14: Remove “(ii)” and restructure the sentence; adding “as well as” may improve readability.

Line 49: Please rephrase “in a narrow region of the nucleus.” The expression “in a small region of the nuclear envelope” may better convey the meaning.

Line 55: The statement “Segregation of homologous chromosomes in diplotene leads to the appearance of a split between the homologs.” is incorrect. In diplotene, homologous chromosomes remain connected.

Line 60: “acrocentric chromosome (or per arm of a dicentric chromosome)” — do you mean a chromosome arm? Please clarify.

Line 100: Replace “delay in chiasmata expression at the diakinesis” with “delay in chiasmata appearance at diakinesis.”

Line 126: “1% osmium tetroxide in the same buffer” — this fixation is usually used for electron microscopy or when membrane preservation is desired. Have you tried fixing preparations without osmium tetroxide?

Line 147: “Sidhu with coworkers” — please consider using “Sidhu et al.”, “Sidhu and colleagues”, or “Sidhu and co-workers.”

Line 150: “the Image Bank of the Individual Chromosomes of L. saxatilis with high-resolution G-banding patterns” — please define Image Bank. Perhaps: “a collection of high-resolution G-banded images of individual L. saxatilis chromosomes.”

Figure 1: “Semithin section of the Littorina saxatilis testis lobule, demonstrating male germline cells at different stages of spermatogenesis” — please provide evidence on how the stages were identified, perhaps by citing your previous studies where similar tissues were shown to correspond to spermatocytes I or II.

Line 178: “Three to four nucleoli are also present, but they were three to four times smaller than in spermatogonia.” — please explain how you differentiated spermatogonia from spermatocytes. Was this distinction based solely on nucleolar size? Please provide evidence that this is a sufficient criterion.

Line 183: “The stages of leptotene and zygotene morphologically can be referred to as mainstream stages of L. saxatilis male meiosis.” — do you mean these stages are the most frequently observed? Please clarify.

Figure 2: I cannot see a clear difference among stages 2F–2H. Please indicate the defining characteristics with arrows or labels.

Line 245: “prekaryosomal, complete karyosomal, postkaryosomal” — please define each stage and indicate the distinguishing features in the figure.

Line 333: “We used the high-resolution G-banding Image Bank of individual mitotic chromosomes (abbr. The Image Bank)” — please define what the Image Bank is and explain how it was created.

Line 524: “the expression of condensed chromatin beads” — please clarify what is meant by “chromatin beads.” How did you demonstrate their expression (transcriptional activity)?

Line 731: “New data may be useful for better understanding reproductive isolation and the evolutionary processes of speciation in the complex of closely related species of the ‘saxatilis’ group of North Atlantic littorinids.” — please elaborate on how unusual meiosis could influence reproductive isolation; this connection is not yet clear.

Comments on the Quality of English Language

There are many non-canonical terms that need clarification or definition

Author Response

Thank you again for critical reading of our MS, your valuable comments and efforts to improve the text of the manuscript. Below are our responses to the criticisms.

Comment 1. Line 14: Remove “(ii)” and restructure the sentence; adding “as well as” may improve readability.

Response 1. Fixed, thank you.

Comment 2. Line 49: Please rephrase “in a narrow region of the nucleus.” The expression “in a small region of the nuclear envelope” may better convey the meaning.

Response 2. Thank you. This sentence has been modified in accordance with your suggestion.

Comment 3. Line 55: The statement “Segregation of homologous chromosomes in diplotene leads to the appearance of a split between the homologs.” is incorrect. In diplotene, homologous chromosomes remain connected.

Response 3. Thank you for this point. We have changed the sentence as follows: “The segregation of homologous chromosomes in diplotene leads to an increase in the distance between homologues. This is most likely occurs due to the partial disassembly of synaptonemal complexes.”

Comment 4. Line 60: “acrocentric chromosome (or per arm of a dicentric chromosome)” — do you mean a chromosome arm? Please clarify.

Response 4. Yes, of course, "chromosome arm." Changes have been made.

Comment 5. Line 100: Replace “delay in chiasmata expression at the diakinesis” with “delay in chiasmata appearance at diakinesis.”

Response 5. Thanks, fixed it.

Comment 6. Line 126: “1% osmium tetroxide in the same buffer” — this fixation is usually used for electron microscopy or when membrane preservation is desired. Have you tried fixing preparations without osmium tetroxide?

Response 6. Yes, we also routinely prepared histological paraffin sections stained with hematoxylin and eosin. However, these appeared to be significantly less representative than semithin sections. We previously presented a detailed definition of the stages of spermatogenesis and spermiogenesis with sufficient resolution [ref. 34], which we now report in the text with an appropriate sentence.

Comment 7. Line 147: “Sidhu with coworkers” — please consider using “Sidhu et al.”, “Sidhu and colleagues”, or “Sidhu and co-workers.”

Response 7. Fixed.

Comment 8. Line 150: “the Image Bank of the Individual Chromosomes of L. saxatilis with high-resolution G-banding patterns” — please define Image Bank. Perhaps: “a collection of high-resolution G-banded images of individual L. saxatilis chromosomes.”

Response 8. Thank you. We have additionally presented the definition of the Image Bank, following your valuable suggestion.

Comment 9. Figure 1: “Semithin section of the Littorina saxatilis testis lobule, demonstrating male germline cells at different stages of spermatogenesis” — please provide evidence on how the stages were identified, perhaps by citing your previous studies where similar tissues were shown to correspond to spermatocytes I or II.

Response 9. Details of the microanatomy of L. saxatilis testes with their unusual lobular-cystic structure, as well as the precise determination of the cellular composition of the testicular lobules with sufficient resolution, were presented in our previous publications [refs. 34 and 40]. In fact, we could exclude Fig. 1 from the MS, but we believe that a general view of the gonad section would be useful to readers.

Comment 10. Line 178: “Three to four nucleoli are also present, but they were three to four times smaller than in spermatogonia.” — please explain how you differentiated spermatogonia from spermatocytes. Was this distinction based solely on nucleolar size? Please provide evidence that this is a sufficient criterion.

Response 10. Yes, the only morphological characteristic of interphase spermatogonia is the markedly enlarged nucleoli. However, there is no evidence that this criterion is sufficient to discriminate such spermatogoniа from interphase primary spermatocytes, so the term “interphase spermatogoniа” has been replaced by “putative interphase spermatogoniа.”

Comment 11. Line 183: “The stages of leptotene and zygotene morphologically can be referred to as mainstream stages of L. saxatilis male meiosis.” — do you mean these stages are the most frequently observed? Please clarify.

Response 11. We mean “canonical stages”, and the corresponding replacement is made in the text.

Comment 12. Figure 2: I cannot see a clear difference among stages 2F–2H. Please indicate the defining characteristics with arrows or labels.

Response 12. The pachytene stage in male meiosis of L. saxatilis can be divided into early, mid and late substages depending on the state of fusion of homologous chromomeres. Early pachytene spermatocytes contain 17 bivalents (Figure 2f), the homologous chromomeres of which are joined by cross-bridges, forming bands. In late pachytene spermatocytes, all bivalents appear as necklace-like structures due to complete fusion of both homologous chromomeres (Figures 2h). Mid-pachytene spreads exhibited different patterns of homologous chromomere fusion, and both banded and necklace-like bivalents are observed (Figure 2g). In this figure (2g), banded bivalents are now marked by arrows.

Comment 13. Line 245: “prekaryosomal, complete karyosomal, postkaryosomal” — please define each stage and indicate the distinguishing features in the figure.

Response 13. We have added the following text fragment explaining the stages of karyosome development:

The pre-karysome, sometimes called the early karyosome/karyosphere [23], reflects the incomplete packaging of all chromosomes in the nucleus into a single “knot” (karyosome). At this substage, a certain chromosome-free space is visible in the nucleus. When the process of chromosome aggregation into a karyosome is complete, this chromatin structure occupies a rather limited space in the nucleus, which corresponds well to the definition of a karyosome [22, 23]. Subsequently, the karyosome subsequently loosens again, and we called this period the post-karyosomal substage, preceding the onset of diakinesis.

Comment 14. Line 333: “We used the high-resolution G-banding Image Bank of individual mitotic chromosomes (abbr. The Image Bank)” — please define what the Image Bank is and explain how it was created.

Response 14. We have changed the beginning of Section 3.3.1 as follows:

We compiled a collection of high-resolution G-band images of individual L. saxatilis chromosomes, termed the Image Bank in this study, which was then used as a primary tool to identify individual bivalents in spreads and to compare G-banding pattern in bi-valents at pachytene, diakinesis, and metaphase I of male meiosis in this species.

Comment 15. Line 524: “the expression of condensed chromatin beads” — please clarify what is meant by “chromatin beads.” How did you demonstrate their expression (transcriptional activity)?

Response 15. We changed this as follows “the appearance of small chromomere-like structures of condensed chromatin”, since pachytene chromomeres will most likely be formed on the basis of these

Comment 16. Line 731: “New data may be useful for better understanding reproductive isolation and the evolutionary processes of speciation in the complex of closely related species of the ‘saxatilis’ group of North Atlantic littorinids.” — please elaborate on how unusual meiosis could influence reproductive isolation; this connection is not yet clear.

Response 16. We have rewritten the Conclusion section to remove any preliminary speculation on this topic.

Comment 17. Comments on the Quality of English Language

There are many non-canonical terms that need clarification or definition

Response 17. Taking into account the unusual features of the male meiosis in L. saxatilis and following the reviewer's suggestions, we have tried to provide additional explanations of non-canonical terms, trying to reduce them to a minimum in the text.